# Chromatin accessibility dynamics and single cell RNA-Seq reveal new regulators of regeneration in neural progenitors

Anneke Dixie Kakebeen[1], Alexander Daniel Chitsazan[2],
Madison Corinne Williams[1], Lauren M Saunders[3], Andrea Elizabeth Wills[1]*

[1]Department of Biochemistry, University of Washington, Seattle, United States;
[2]Oregon Health Sciences Center For Early Detection Advanced Research Center
(CEDAR), Portland, United States; [3]Department of Genome Sciences, University of
Washington, Seattle, United States

**Abstract** Vertebrate appendage regeneration requires precisely coordinated remodeling of the transcriptional landscape to enable the growth and differentiation of new tissue, a process executed over multiple days and across dozens of cell types. The heterogeneity of tissues and temporally-sensitive fate decisions involved has made it difficult to articulate the gene regulatory programs enabling regeneration of individual cell types. To better understand how a regenerative program is fulfilled by neural progenitor cells (NPCs) of the spinal cord, we analyzed *pax6*-expressing NPCs isolated from regenerating *Xenopus tropicalis* tails. By intersecting chromatin accessibility data with single-cell transcriptomics, we find that NPCs place an early priority on neuronal differentiation. Late in regeneration, the priority returns to proliferation. Our analyses identify Pbx3 and Meis1 as critical regulators of tail regeneration and axon organization. Overall, we use transcriptional regulatory dynamics to present a new model for cell fate decisions and their regulators in NPCs during regeneration.

*For correspondence:
aewills@uw.edu

Competing interests: The
authors declare that no
competing interests exist.

Reviewing editor: Marianne E
Bronner, California Institute of
Technology, United States

## Introduction

In contrast to mammals, *Xenopus* tadpoles are able to undergo scarless healing and full regeneration of the limb, spinal cord, or tail after injury (*Beck et al., 2009*; *Kakebeen and Wills, 2019*; *Lee-Liu et al., 2017*; *Tseng and Levin, 2008*). While lifelong regenerative healing is a characteristic shared by many amphibians and fish, the regenerative capacity of *Xenopus* declines during metamorphosis, and is lost in the adult (*Filoni and Bosco, 1981*; *Mitogawa et al., 2015*; *Suzuki et al., 2006*). *Xenopus* therefore represents an especially useful model for understanding the cell-intrinsic and –extrinsic properties governing regeneration.

In *Xenopus* as in other regenerative animals, the event of a major injury triggers a rapid transcriptional remodeling of the injured tissue. It is now well-established that some aspects of this remodeling recapitulate developmental signaling events. In particular, developmental signaling pathways such as Wnt, FGF, BMP, TGF-ß, Notch and Shh are upregulated, and are required for full regeneration of the limb, tail, and spinal cord (*Beck et al., 2003*; *Ho and Whitman, 2008*; *Slack et al., 2008*; *Taniguchi et al., 2014*). Genome-wide transcriptomic studies have confirmed that numerous genes associated with embryonic development are re-expressed during regeneration (*Chang et al., 2017*; *Lee-Liu et al., 2014*; *Love et al., 2011*). However, these studies have been carried out on bulk regenerating tissue, making it difficult to identify what signals or factors are required to promote regeneration in specific cell types. Recently, single-cell transcriptomic analysis (scRNA-Seq) of both the regenerating *Xenopus laevis* tail and the regenerating axolotl limb have begun to identify the transcriptional signatures associated with distinct cell types (*Aztekin et al., 2019*; *Gerber et al.,*

*2018*; *Pelzer et al., 2020*). These studies also highlighted intriguing distinctions between the models. The regenerating axolotl limb shows a transcriptional convergence between all connective tissue cell types, associated with the formation of the undifferentiated limb blastema (*Gerber et al., 2018*), while in the *Xenopus* tail, cell type identities are kept clearly intact as regeneration progresses (*Aztekin et al., 2019*). While these studies substantially advance our atlas of the genes expressed by different cell types in regeneration, they do not identify the transcription factors that interpret and respond to injury cues, nor the transcriptional regulatory elements that trigger changes in expression profiles.

To address these gaps, we sought to articulate the gene regulatory network and transcriptional dynamics associated with regeneration by specifically targeting NPCs, a critical cell type for regeneration. NPCs represent the focus of some therapeutic efforts to restore human motor function following spinal cord injury in regenerative medicine (*Khazaei et al., 2019*; *Levi et al., 2018*; *Shin et al., 2015*) and much effort has been applied to defining the in vivo and in vitro programs that guide their cell fate decisions in the developing spinal cord as well as in culture. During development, neural stem cells give rise to distinct domains of lineage-restricted progenitor cells defined by their expression of specific transcription factors across the dorsal ventral axis (*Alaynick et al., 2011*; *Lai et al., 2016*). The decision to exit the cell cycle and undergo terminal neuronal differentiation is directed by extrinsic cues, particularly repression of Notch signaling, which is highly conserved across vertebrates (*Lara-Ramirez et al., 2019*). Following amputation, the spinal cord first begins to repair by closing the open end to form the tube-like neural ampulla, a structure formed by ependymal cells. Progenitor cells then contribute to both growth and new differentiation as the newly regenerated spinal cord elongates (*Beck et al., 2009*; *Gaete et al., 2012*; *Stefanelli, 1951*). Therefore, neural progenitors must balance their decisions between proliferating to repopulate lost tissue and differentiating to mature neurons. The global transcriptional dynamics that govern these decisions, including the transcription factors required and their target sites across the genome, remain poorly defined.

To identify gene regulatory dynamics in neural progenitors during regeneration, we analyzed the transcription and chromatin accessibility profiles of neural progenitors purified from *pax6:GFP* transgenic *Xenopus tropicalis* tails over a regenerative timecourse. Using both ATAC-Seq and scRNA-Seq, we show that by 24 hr post amputation (hpa), *pax6+* neural progenitors place a high priority on differentiation to motor and interneuron subtypes, and only later at 72hpa do they accessibilize genes associated with proliferation and self-renewal. By intersecting transcriptional and chromatin accessibility data, we identified candidate transcriptional regulators associated with both early and later changes in this neural progenitor gene expression profile. Further analysis of two of these, Pbx3 and Meis1, revealed similar loss-of-function phenotypes, including reduced tail regeneration and disorganization of regenerated neural tissues. Taken together, we present an integrated transcriptional regulatory analysis of regeneration in *pax6*-expressing NPCs that highlights both parallels and distinctions relative to embryonic development, and present novel candidates regulating regeneration in this critical cell type.

## Results

### Transgenic *pax6:GFP* is expressed in NPCs during regeneration

We selected the *Xtr.Tg(pax6:GFP;cryga:RFP;actc1:RFP)* transgenic line (hereafter abbreviated *pax6:GFP*) to track NPCs over the course of regeneration (*Hartley et al., 2001*; *Hirsch et al., 2002*; *Horb et al., 2019*; 31,88). *Pax6* is a highly conserved paired box transcription factor essential for central nervous system development (*Bel-Vialar et al., 2007*; *Osumi et al., 2008*; *Walther and Gruss, 1991*). Because the expression pattern of this line has primarily been characterized earlier in development, we first wanted to confirm that GFP was found in a pattern consistent with *pax6* expression and NPC distribution (*Ericson et al., 1997*; *Hartley et al., 2001*). To do so, we imaged whole mount tadpoles to identify where the *pax6:GFP+* cells were localized anteroposteriorly in a stage 41 tadpole, finding that *pax6* is expressed in the forebrain, hindbrain, eye, and spinal cord (*Figure 1A*). In lateral confocal images of whole-mount transgenic tails, we found that the GFP expression domain spanned the same dorsal ventral domain as the spinal cord (*Figure 1B*). Transverse sections of a stage 41 tadpole also revealed a broad dorsoventral *pax6+* domain surrounding

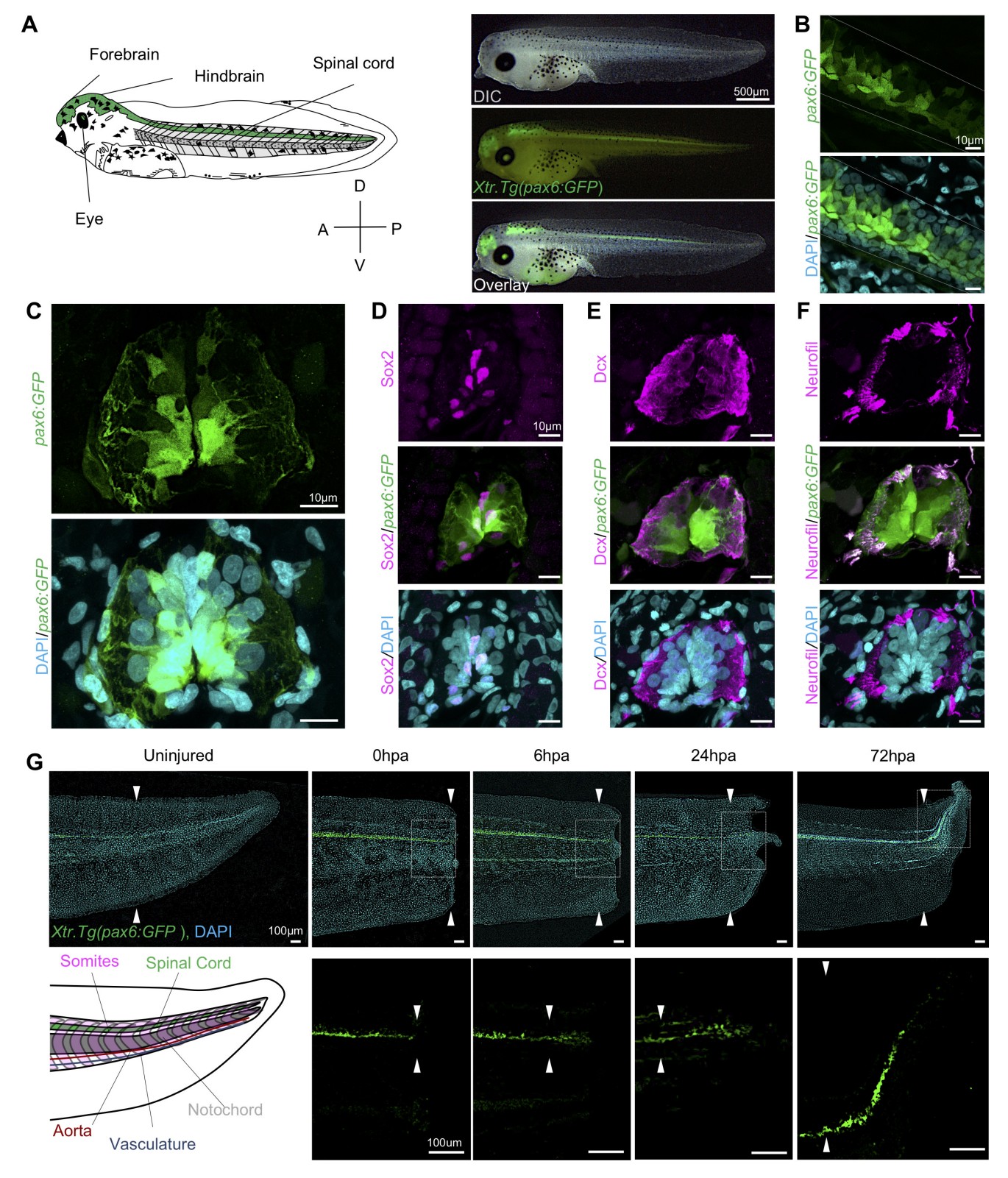

**Figure 1.** *Expression dynamics of pax6:GFP in the regenerating spinal cord.* (**A**) Left, cartoon depiction of a NF stage 41 tadpoles with known *pax6*+ domains colored green. Right, Stage 41, transgenic tadpole expressing GFP under the control of *pax6* promoter tadpole. For all images in this figure, the green channel represents *pax6* reporter GFP fluorescence and the cyan channel represents DAPI staining. (**B**) Confocal image of a lateral view of a whole-mount transgenic stage 41 tadpole. (**C–F**) Transverse cryosections through the posterior spinal cord of a transgenic stage 41 tadpole. (**D–F**)
*Figure 1 continued on next page*

*Figure 1 continued*

Immunofluorescence images of the spinal cord sections stained with (D) anti-Sox2, (E) anti-Dcx, and (F) anti-neurofilament. (G) Regeneration timecourse of Xtr.Tg(*pax6:GFP*) tadpole over the first 72 hr following tail amputation. The white box in the top photos correspond with enlarged green channel below. White arrows indicate amputation plane.

The online version of this article includes the following figure supplement(s) for figure 1:

**Figure supplement 1.** FACS gating was set using wild-type tadpoles to avoid false-positive GFP+ cells.

the central canal of the spinal cord (*Figure 1C*). GFP expression in transgenic tadpoles included much of the same domain as the NSC marker Sox2 (*Figure 1D*). Notably, both nuclear Sox2 and cytoplasmic GFP driven by *pax6* are localized medially around the central canal, while markers associated with differentiated neurons such as Dcx and neurofilament are largely found in the spinal cord periphery (*Figure 1E,F*). Thus we concluded that we could use GFP expression to capture a large proportion of NPCs and thereby interrogate their population-level regulatory dynamics, although we can't conclusively rule out the possibility that some NSCs are excluded from this domain, or that some differentiated neurons may be included in GFP-expressing cells.

To better understand when NPCs enter the regenerating spinal cord, we amputated *pax6:GFP* transgenic tadpoles and followed reentry of GFP+ cells into the regenerating spinal cord tissue. Transgenic tadpoles were collected at 0 hr post amputation (0hpa), 6hpa, 24hpa, and 72hpa and imaged for reporter GFP fluorescence. We found that *pax6:GFP+* cells can be clearly identified in the regenerating tail as early as 6hpa (*Figure 1G*). By 24hpa, the GFP+ cells are found in the bulb of the neural ampulla (*Beck et al., 2009*; *Stefanelli, 1951*). By 72hpa, GFP+ cells are present all along the anteroposterior axis of the regenerated spinal cord. From these experiments, we concluded that *pax6:GFP+* cells are appropriately localized as they repopulate the regenerated spinal cord.

## The chromatin accessibility profile of FACS-sorted NPCs is highly enriched for neural-specific regulatory regions

We set out to identify the chromatin accessibility signature of NPCs over a regeneration timecourse using ATAC-Seq (*Buenrostro et al., 2013*). To perform this cell-type specific genomic analysis of regeneration in neural progenitors, we optimized a method to isolate *pax6:GFP+* fluorescent cells from transgenic tadpoles by flow cytometry (*Figure 1—figure supplement 1*) (see Materials and methods for details). The posterior third of stage 41 tadpole tails were amputated and ATAC-Seq libraries were made from amputated tail tips (uninjured) or the newly regenerated tissue at 0hpa, 6hpa, 24hpa, or 72hpa (*Figure 2A*). The timepoints were collected in two conditions. The first, 'all-tissue' libraries refer to libraries made from total tail tissue, which includes all of the cell types in the tail. The second, '*pax6*' refers to libraries made from GFP+ sorted cells (*Figure 2A*). Each library was prepared from between 1200 and 4000 cells, either with no GFP gate applied and therefore containing all cell types ('all tissue') or with a GFP gate applied as in *Figure 1—figure supplement 1C* ('*pax6*'). Sample preparation details for each library are detailed in *Supplementary file 1a*. Quantitative RT-PCR of sorted GFP+ cells confirmed highly elevated expression of *gfp* and *pax6*, with low levels of neuronal-specific *tubb2b* or *cardiac actin*, confirming the identity of these cells with low levels of contamination (*Figure 1—figure supplement 1D*). Libraries for each timepoint and condition were made in triplicate, multiplexed and sequenced on the Illumina Next-Seq platform (see Materials and methods for details). Read alignment and peak calling was performed using an inhouse pipeline detailed in Materials and methods; library QC metrics are reported in *Supplementary file 1b* and *Figure 2—figure supplement 1*. Sequenced libraries were selected for analysis if they met minimum ENCODE standards for ATAC-Seq libraries (www.encodeproject.org/atac-seq).

Having verified the quality of ATAC-Seq libraries made from sorted *pax6:GFP+* cells and bulk tail tissue, we first confirmed that the overall tissue conditions had chromatin accessibility signatures that could be easily differentiated. Taking an unbiased approach, we used multi-dimensional scaling (MDS) to identify how similar or different the *pax6* and all-tissue libraries are overall. We found that the *pax6* libraries cluster away from the clustered all-tissue libraries (*Figure 2B*). We then examined individual gene regulatory loci. The majority of peaks called were located within 500 bp of a transcription start site for both all tissue and *pax6* libraries, confirming that the majority of ATAC-Seq

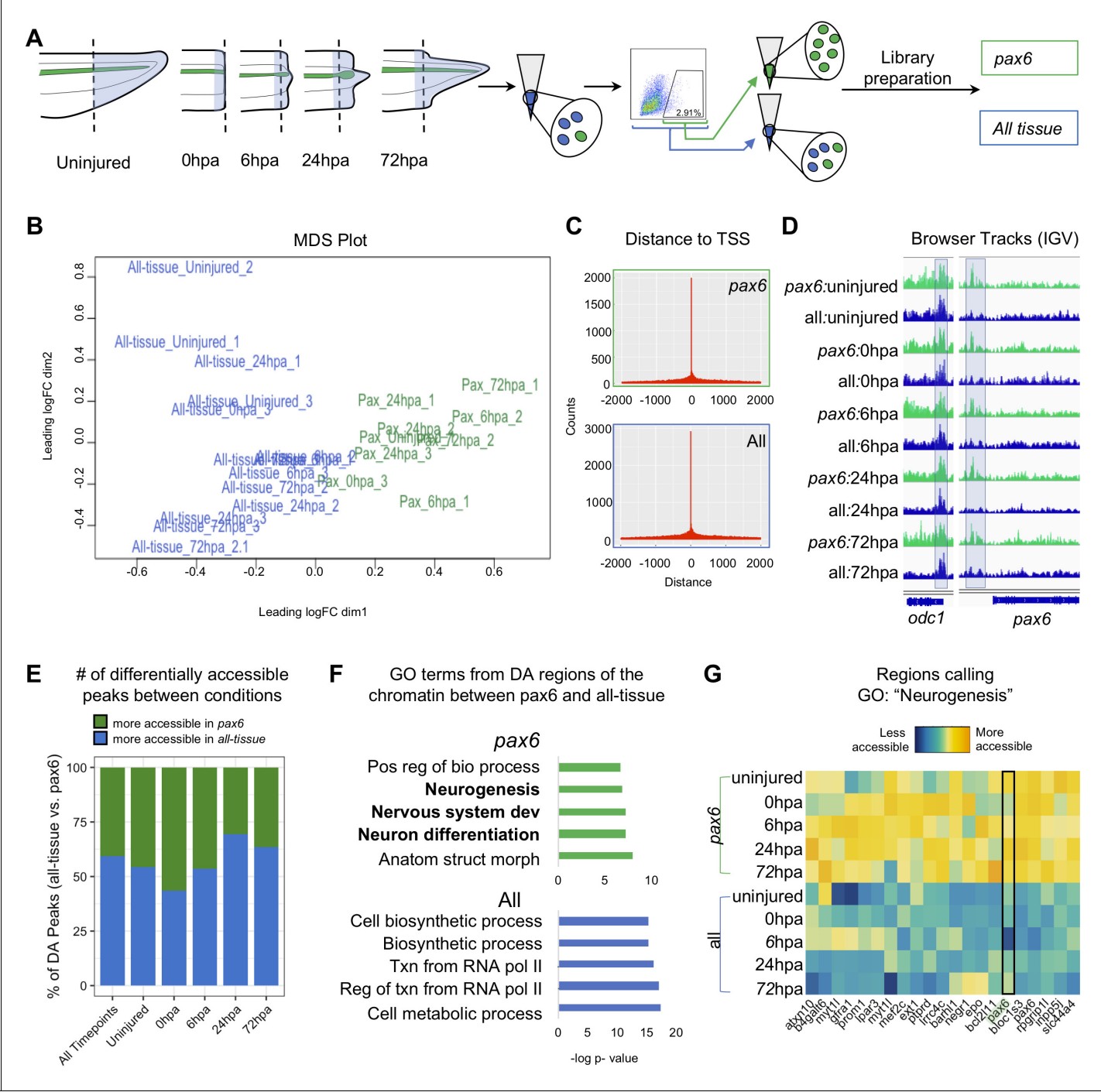

**Figure 2.** *Pax6 ATAC-Seq libraries resolve accessible neural specific regions that were not identified in all-tissue libraries.* (A) Experimental design for FACS isolation of reporter cells for sequencing library preparations. (B) MDS plot of sequenced *pax6* and all-tissue ATAC-Seq libraries. (C) Representative histograms showing distance of called peaks to transcription start sites (tss). (D) IGV browser tracks of a house keeping gene, *odc* and reporter line specific gene, *pax6*. Shaded bar in *odc* represents a peak retained in *pax6* and all-tissue libraries; shaded bar in *pax6* represents a peak called in *pax6* libraries, but not in all-tissue. (E) Bar plot showing percent of differentially accessible (DA) peaks that are more accessible in *pax6* or all-tissue libraries within an individual timepoint or across all timepoints. (F) Differentially accessible regions were annotated to the nearest TSS and used to call GO:BP terms. Top, GO terms from regions more accessible in *pax6* libraries than in all-tissue libraries. Bottom, GO terms from regions more accessible in all-tissue libraries than *pax6* libraries. (G) Heat map depicting accessibility of 21/254 randomly sampled peaks that call the GO term: Neurogenesis. *pax6* promoter region is boxed.

The online version of this article includes the following figure supplement(s) for figure 2:

*Figure 2 continued on next page*

*Figure 2 continued*

**Figure supplement 1.** Low input ATAC-Seq libraries meet sufficient quality for downstream analysis.

signal is concentrated in nucleosome-depleted promoters, as expected (*Buenrostro et al., 2013*; *Figure 2C*). Both *pax6* and all-tissue libraries shared common peaks around a control housekeeping promoter (*odc*) (*Dhorne-Pollet et al., 2013*) without any apparent significant differences in accessibility (*Figure 2D*, left). By contrast, upstream of the *pax6* TSS, all *pax6* library timepoints share a peak that is absent in all-tissue libraries (*Figure 2D*, right). From this first proof-of-principle test, we concluded that ATAC-Seq libraries from sorted cells can be used to identify chromatin accessibility differences in NPCs relative to other tissues, and that ATAC-Seq of these sorted cells can be used to find tissue-specific accessible regions that are beneath the detection sensitivity of bulk analysis of all of the tissues together. We then progressed to genome-wide analysis of differences between *pax6: GFP*+ calls and bulk tissue.

Our next goal was to identify categories of gene regulatory regions that have increased accessibility in NPCs relative to bulk tail tissue. To this end, we interrogated the global NPC chromatin signature by compiling all *pax6* library peaks into one peak file and all of the all-tissue peaks into a second peak file and calling differentially accessible peaks between these two compiled files. We identified 3604 regions that were more accessible in the *pax6* libraries than the all-tissue libraries, which we assigned to the nearest TSS, and used the associated gene names as input for gene ontology (GO) analysis using gProfileR2 (*Raudvere et al., 2019*; *Figure 2E–F*). Genes with increased accessibility in *pax6:GFP*+ cells showed statistically significant enrichment for neural-associated terms such as 'Neurogenesis', 'Nervous system development', and 'Neuron differentiation,' confirming the neural progenitor identity of our *pax6:GFP* cells (*Figure 2F*, top, and *Supplementary file 1c*). We visually confirmed the increased accessibility of regions calling the term 'Neurogenesis' with a heat map (*Figure 2G*). The reverse analysis identified 5251 regions that are more accessible in all-tissue libraries than *pax6* libraries, which are associated with broad GO categories relating to ubiquitous cellular processes (i.e. 'Cellular biosynthetic process', 'Transcription from RNA Polymerase II', and Cellular metabolic process') (*Figure 2F*, bottom, *Supplementary file 1d*). From this analysis, we concluded that the overall chromatin accessibility of NPCs is readily distinguished from bulk tissue and strongly reflects their distinct neural character.

## Analysis of differentially accessible peaks between *pax6* libraries reveals an early prioritization of neuronal differentiation in regeneration

Having established that the overall chromatin signature of NPCs could be used to identify neural-specific processes, we next identified the cellular processes being prioritized by NPCs at discrete regenerative stages. We therefore systematically identified differentially accessible regions of the chromatin that were uniquely accessible at each timepoint relative to the flanking timepoints and performed GO analysis on the genes neighboring these regions (*Figure 3A–C*). GO analysis returned ~30–100 terms, thus, to distill the list to main processes, we used ReviGO's semantics algorithm to reduce redundancy of the GO terms (*Supek et al., 2011*). *Supplementary files 1e-g* contain the entire list of GO terms returned from gProfiler and *Supplementary files 1h-j* contain the results from ReviGO.

We first examined genes associated with regulatory regions that are uniquely accessible at 6hpa. Genes preferentially accessible at 6hpa called six families of GO terms (*Figure 3D*). Notably, 'Nephron Tubule Morphogenesis' was well represented, and included regulatory regions from *foxd1*, *gata3*, *smo*, *sall1*, *osr1*, *pax2*, *stat1*, *hoxd11*, *wnt6*, *nog*, *wnt11*, *irx3*, *cited1* (*Figure 3G*). In addition to patterning the nephron, the majority of these genes are expressed in neural stem cells during dorsoventral neural patterning of the spinal cord and early differentiation (*Alaynick et al., 2011*; *Delile et al., 2019*). At this time in regeneration, we observe the spinal cord repairing itself by closing the neural tube and forming the neural ampulla. We have shown in *Figure 1D* that pax6:*GFP*+ cells populate the regenerated neural ampulla, therefore, we hypothesize that these cells represent ependymoglial cells with NSC character that have been described previously (*Beck et al., 2009*;

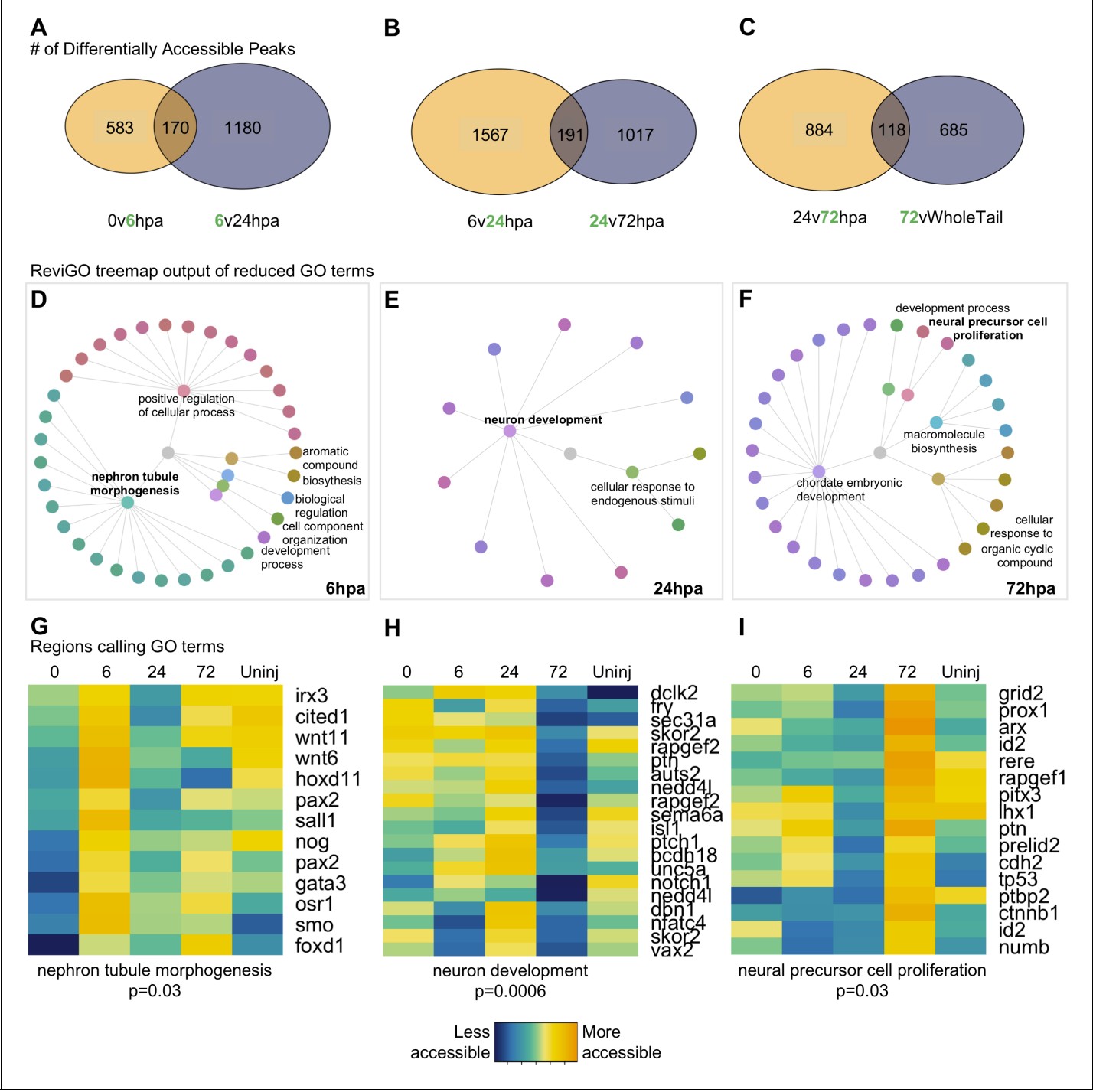

**Figure 3.** Differential accessibility analysis of pax6 libraries over regenerative time reveals chromatin accessibility prioritizes first tubule morphogenesis, followed by neural differentiation, and later proliferation. From left to right, data represents the 6hpa timepoint, 24hpa timepoint, and 72hpa timepoint. (A–C) Venn diagrams depicting number of unique and shared peaks called in each of the timepoint contrasts within the *pax6* libraries. The intersect represents peaks that were both becoming accessible at a given timepoint, then losing accessibility. The green number in each contrast represents the timepoint that is more accessible. (D–F) All regions represented in (A–C) were used for GO:BP analysis with gProfileR2. ReviGO was then used to reduce redundancy in output GO terms and identify main families of terms. Tree graphs depict the hierarchy of reduced GO list. The central grey circle is the vertex of the graph representing the set of all GO terms included in the data, the second level of circles are the families of GO terms called by ReviGO, and the third level of circles represents GO terms in each family. Terms provided are the family names. (G–I) Heatmap of accessibility of regions that were used to call GO terms: 'Nephron Tubule Morphogenesis', 'Neuron Differentiation', and 'Neural Precursor Cell Proliferation'.

*Chernoff et al., 2018*; *McHedlishvili et al., 2007*; *Muñoz et al., 2015*; *O'Hara et al., 1992*; *Reimer et al., 2009*).

We then identified regions that were most accessible at the 24hpa timepoint. This analysis recovered two main families of GO terms, the largest of which was 'Neuron Development' (*Figure 3E*). Strikingly, the majority of genes that call this GO term become more accessible from 6 to 24hpa, but lose that accessibility at 72hpa (*Figure 3H*). Genes with preferentially accessible regulatory regions at 24hpa include factors known to function in multiple aspects of neuronal differentiation and growth. Examples of these include neuritogenesis and growth cone formation (*dclk2, rapgef2, auts2, nedd4l, unc5a*), axonogenesis and axon guidance (*dbn1, sema6a, fry, ptch1*), neuronal migration (*dclk2, rapgef2, auts2, pcdh18*), and transcriptional regulation of neuron differentiation (*skor2, isl1, nfatc4, vax2*). The enrichment of these terms, and their specificity for functions carried out by differentiating neurons rather than proliferating progenitors, agrees well with previous published bulk RNA-Seq analysis (*Chang et al., 2017*) showing enrichment for similar neuronal morphogenesis terms, such as axonogenesis and dendritogenesis, in the first day post amputation.

Finally, we assessed the processes being prioritized at 72hpa. This analysis resulted in five main GO families being called (*Figure 3F*), including 'neuronal precursor cell proliferation'. Notably, cell cycle and proliferation related terms were absent at the 6hpa and 24hpa timepoints. The regions calling this term were all more accessible at 72hpa than the 24hpa timepoint, however about half were not differentially accessible with respect to the uninjured tail (*Figure 3I*), suggesting that by 72hpa, these regions are regaining similarity to their uninjured state. Altogether, we were able to resolve processes specific to neural progenitor cell function at each timepoint. These suggest that NPCs first prioritize ependymal tube morphogenesis at 6hpa, neural differentiation at 24hpa, and cell proliferation at 72hpa.

## Comparative analysis of uninjured and 24hpa tail single cell RNA-Seq data sets reveals expansion in differentiated neuronal clusters at 24hpa

Our ATAC-Seq analysis suggested that neural progenitor cells may place a high transcriptional regulatory priority on neuronal differentiation early in regeneration, at 24hpa. To determine whether this was reflected in the transcriptional profile and cell-type composition of neural cells, we used single-cell RNA-Seq to interrogate the profile of neural cell types present before and 24 hr after amputation. We sequenced the transcriptomes of individual NPCs from uninjured and regenerating tails, an analogous population to those collected for ATAC-seq. In total, we sequenced 2617 and 1,090 cells from uninjured and 24hpa tissues, respectively. Using Seurat we aligned the two data sets with their integrated analyses and used UMAP dimensional reduction to find clusters (*Becht et al., 2018*; *Butler et al., 2018*; *Stuart et al., 2019*). From this analysis, we revealed 19 unique clusters (*Figure 4—figure supplement 1A*), which were enriched for but not fully restricted to the neural lineage. A total of 781 uninjured cells and 296 cells from 24hpa cells fell into neural clusters, which comprised NPC/NSCs (defined by expression of *sox2*), transitioning neurons (*neurog1; neurod1*), and differentiated neurons (*elavl4*) (*Figure 4—figure supplement 1B*). To gain the highest-possible resolution of cell types within these broad neural clusters, we subsetted neural cells, re-performed dimensionality reduction and called eight unique sub-clusters, including spinal cord progenitors, floor plate progenitors, differentiating neurons, interneurons, motor neurons, vulnerable motor neurons, motor neurons (*leptin*+), and dopaminergic neurons (*Figure 4A*). These clusters agreed well with the neural cell types recently identified for tail regeneration in *X. laevis*, and we applied the same naming conventions (*Aztekin et al., 2019*).

Several landmark insights emerged from this single-cell analysis of neural lineages in the tadpole tail. First, we were able to identify new candidate molecular markers for progenitors (*vwcx2l.2, iqca1, daw1*), floor plate progenitors (*bicc, vasn*), differentiating neurons (*nhlh1, tmem178.2*), motor neurons (*prdm14, prdm1, kcnb2*), leptin+ motor neurons (*abcb1, asns, lepr*), vulnerable motor neurons (*mmp17, slc18a3, colq-like*), interneurons (*cnrip1, slc17a7*), and dopaminergic neurons (*pkd2l1, pkd1l2, slc32a1*) (*Figure 4B*). The 'vulnerable motor neurons' term refers to vulnerability of specific neurons to degeneration in ALS (*Kline et al., 2017*). Representative genes for each cell type were identified by using differential expression analysis between cell clusters. Second, we found that cell type identities could be clearly assigned to these clusters at 24 hr after injury (*Figure 4C*). Unlike the connective tissues in the axolotl limb, which lose cell heterogeneity of gene expression after injury

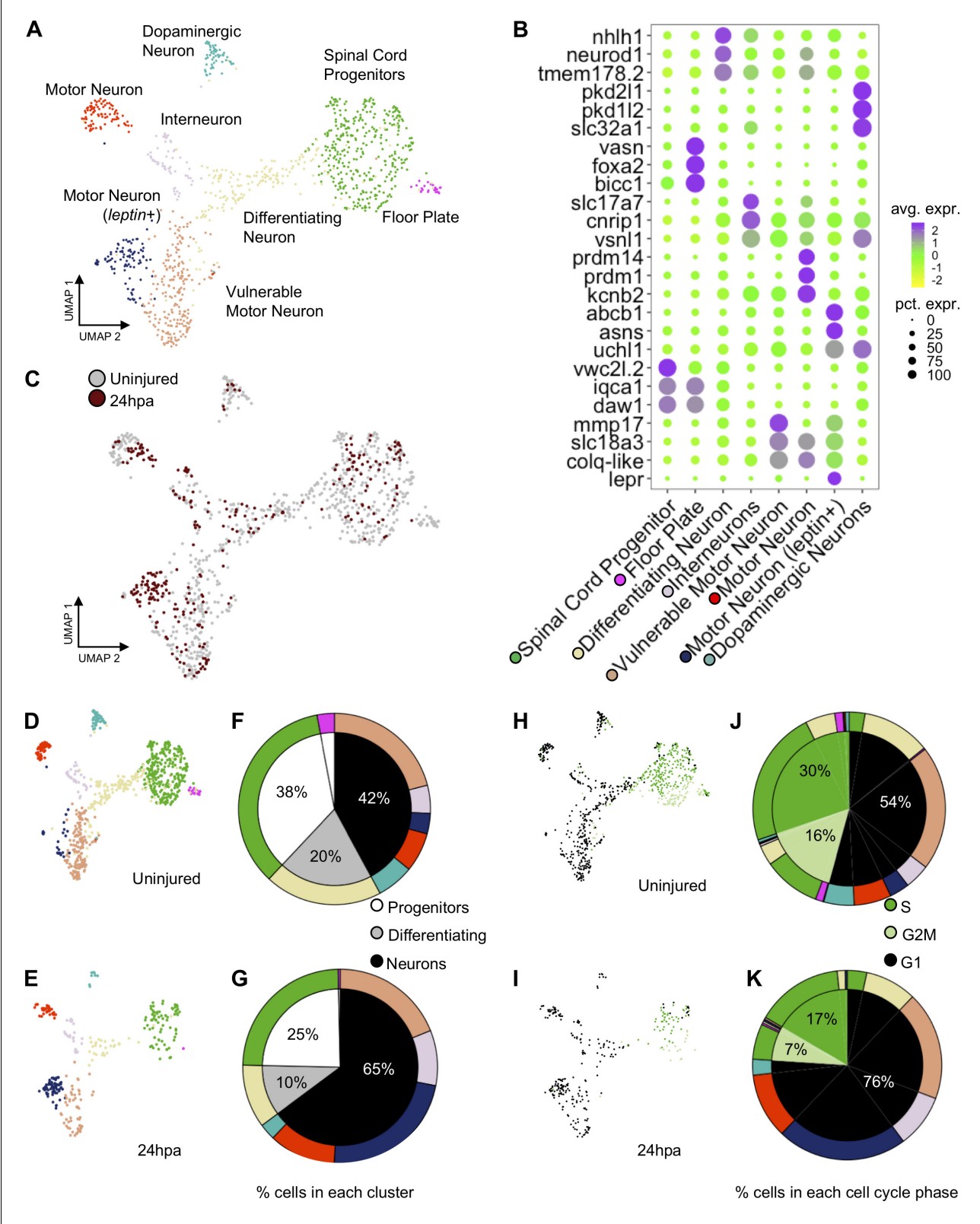

**Figure 4.** *scRNA-Seq of uninjured and 24hpa tails reveals a transcriptional shift to differentiated neuronal types at 24hpa.* (**A**) UMAP projection of integrated uninjured and 24hpa neural lineage cells. seven distinct clusters were identified. (**B**) Dot plot of genes identified as differentially expressed between cell clusters. The top three genes exclusively expressed in each of the seven clusters are shown. Color of circle denotes average gene expression across one cluster and size of circle represents percent of cells in each cluster expressing each gene. (**C**) UMAP projection of neural cells

*Figure 4 continued on next page*

*Figure 4 continued*

colored by timepoint in the uninjured tails and 24hpa tails. UMAP projections of cell clusters split by timepoint to (D) uninjured and (E) 24hpa. (F–G) Sunburst diagrams showing (inside) the percentage of neural cells that are broadly stem cells, differentiating cells, or differentiated neurons and (outside) percentage of neural cells in each of the seven cell clusters called. Colors on outside of the sunburst chart correlate to cluster colors in (A/B). (H–I) UMAP projections of neural clusters colored by predicted cell cycle phase in the (H) uninjured tail and (I) 24hpa tail. (J–K) Sunburst diagrams showing the percentage of all neural cells predicted to be in each cell cycle phase (inside) and the percentage of each cell cluster predicted to be in each cell cycle phase (outside). (J) represents the distribution in the uninjured tail and (K) represents the distribution in the 24hpa tail. Colors on outside of the sunburst chart correlate to cluster colors in (A/B).

The online version of this article includes the following figure supplement(s) for figure 4:

**Figure supplement 1.** scRNA-Seq analysis of all cells sequenced in uninjured and 24hpa tails.

(*Gerber et al., 2018*), neural cells in the 24hpa *Xenopus* spinal cord maintain heterogeneous gene expression.

We next used our scRNA-Seq data to interrogate the hypothesis that neural lineages put an early priority on differentiation rather than proliferation. To address this question and to identify which neuronal types, if any, are being produced, we asked how the relative quantities of cell populations change following injury. To this end, we identified what percent of the total neural population is made up of each cluster under each condition. If neural lineages prioritize differentiation early in regeneration, we would expect to see a decline in proliferating progenitor cells and an increase in one or more differentiated neuronal types. Indeed, at 24hpa, we observe a decrease in the percent of NPCs and differentiating neurons with respect to the uninjured cells (*Figure 4D–G*). In agreement with our GO analysis of 24hpa ATAC-Seq data, we see an expansion in differentiated neuronal clusters including interneurons, vulnerable motor neurons, and motor neurons (*leptin*+) at 24hpa. Overall, NSCs are reduced from 38% to 25%, differentiating neurons are reduced from 20% to 10%, and differentiated neurons increase from 42% to 65%. As previously described, we noted that motor neurons (*leptin*+) are virtually absent in the uninjured tail but are abundant at 24hpa (*Aztekin et al., 2019*).

These transitions in cell type composition are consistent with the hypothesis that neuronal differentiation, especially to specific motor neuron types, is prioritized early in regeneration, with a decline in the relative abundance of proliferative cells.

Our hypothesis is that neuronal differentiation is prioritized early in regeneration while proliferation is decreased. To further test this hypothesis, we asked if we could see a shift in the transcriptomically defined cell cycle state accompanying the change in cell type representation. To this end we used the Seurat cell cycle phase predictor to predict the cell cycle phase of each cell in the neural lineage. In both the uninjured tail and at 24hpa, the NPCs are primarily predicted to be in S and G2M phases, the differentiating neurons are a mix of G2M and G1 phase, and the subtypes of differentiated neurons are primarily made up of G1 phase (*Figure 4H/I*). The uninjured tail neural lineage has 30% cells predicted in S phase, 16% cells predicted in G2M phase, and 54% cells predicted in G1 phase (*Figure 5H/J*). This distribution shifts markedly at the 24hpa timepoint, where the S phase falls to 17%, G2M falls to 7% and G1 increases to 76% (*Figure 4I/K*). Thus, at the 24hpa timepoint, there is a relative increase in the proportion of cells predicted to be in G1 phase and decrease in cells in G2/M/S phases. This correlates with the relative increase in differentiated neurons and decrease in spinal cord progenitors and differentiating cells and supports the hypothesis that NPCs are undergoing differentiation.

We validated expression of several differentiated neuronal genes by in situ hybridization (ISH). We examined *l1cam*, *uchl1*, *nsg1* and *ass1* as candidate neuronal differentiation markers. All these genes are strongly expressed in one or more differentiated neuronal populations but are lowly expressed or absent in NPCs and non-neural cell types by scRNA-SEQ (*Figure 5A–C,G–I,M–O,S–U*). When examined by ISH all four genes are most strongly and distinctly expressed in the neural ampulla at 24hpa (*Figure 5E,K,Q,W*). As a further test of the hypothesis that proliferation is not prioritized until 72hpa, we used phospho-histone H3 (PH3) staining to quantify mitotic cells in optical longitudinal sections of the spinal cord that overlap with the *pax6* domain, finding that proliferative cells are significantly more abundant at 72hpa relative to 24hpa (*Figure 5Y*-AA). These findings further supported our model that differentiation is prioritized at 24hpa, and proliferation at 72hpa.

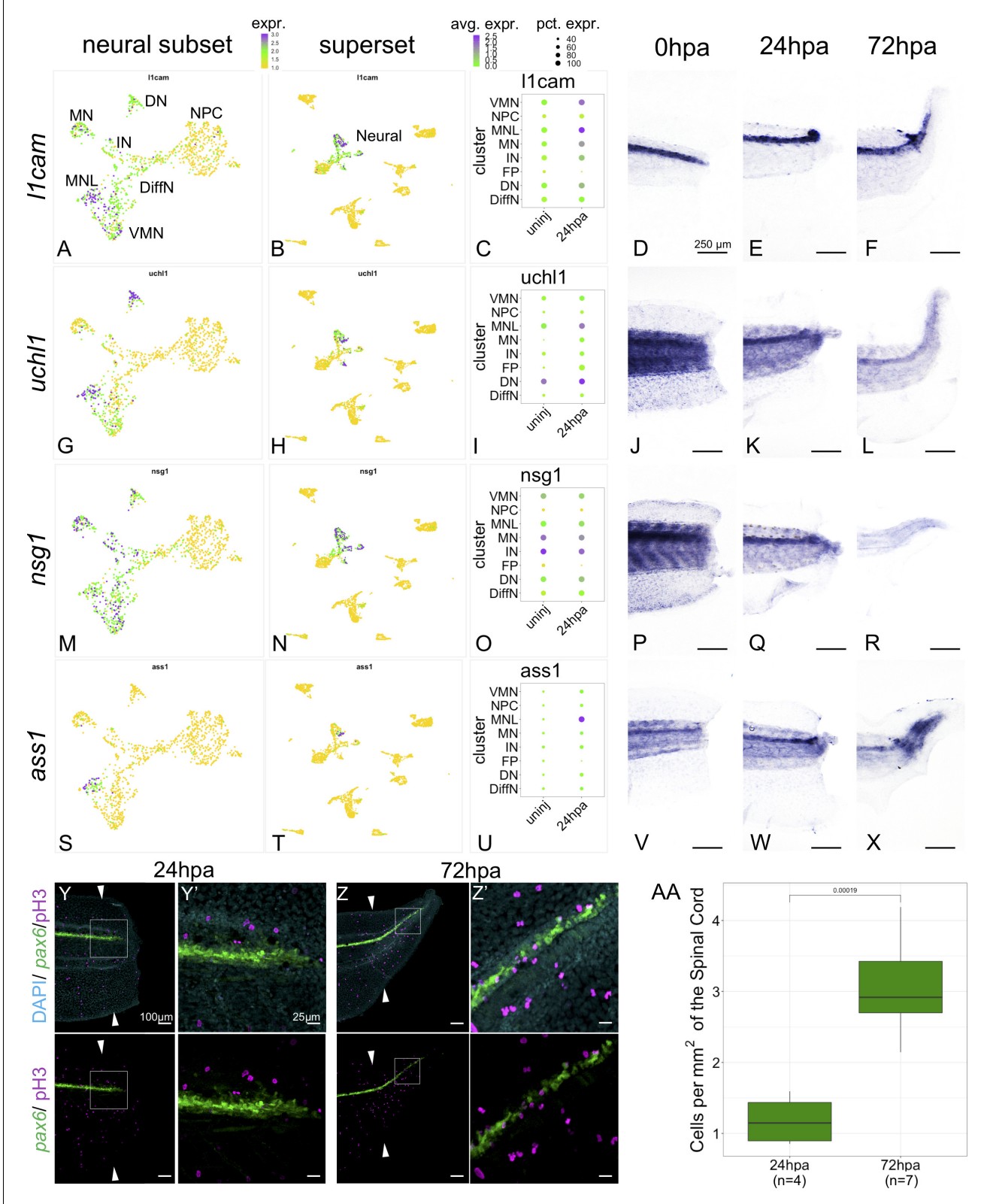

**Figure 5.** *Markers of neuronal differentiation are increased at 24hpa and proliferation increases at 72hpa.* UMAP plots of expression in the neural restricted subset of scRNA-Seq for (A) *l1cam*, (G) *uchl1*, (M) *nsg1*, and (s) *ass1*. UMAP plots of expression in the superset scRNA-Seq dataset for (B) *l1cam*, (H) *uchl1*, (N) *nsg1*, and (T) *ass1*. Note that the topology of neural cell clusters is slightly different when subset and reclustered to give high neural cluster resolution (A,G,M,S) than when neural cells are analyzed together with all other cell types (E,K, Q, W), but the same cells and expression
*Figure 5 continued on next page*

*Figure 5 continued*

data are included in both cases. Dot plots representing average expression of (**C**) *l1cam*, (**I**) *uchl1*, (**O**) *nsg1*, and (**U**) *ass1* per cluster and timepoint. In situ hybridization at 0hpa, 24hpa, and 72hpa for (**D–F**) *l1cam*, (**J–L**) *uchl1*, (**P–R**) *nsg1*, and (**V–X**) *ass1*. (**Y–Z'**) Representative images of 24hpa (**Y/Y'**) and 72hpa (**Z/Z'**) regenerated tails from a *pax6:GFP* transgenic tadpole stained for DAPI (cyan) and mitotic cells with phospho-histone3 (magenta). The white arrows indicate regeneration plane. (**Y'**) and (**Z'**) are enlarged images of the boxed areas in (**Y**) and (**Z**). (**AA**) Boxplot representing the number of cells per regenerated spinal cord area (mm$^2$) at 24hpa and 72hpa. Statistics represent t-test performed between the two timepoints. Abbreviations: Neural Progenitor Cell (NPC), Differentiating Neuron (Diff), Interneuron (IN), Vulnerable Motor Neuron (VMN), Dopaminergic Neuron (DN), Motor Neuron Leptin+ (MNL), Motor Neuron (MN)).

The online version of this article includes the following source data for figure 5:

**Source data 1.** PH3 Cell Count and Regenerated Spinal Cord Area.

## Gene regulatory network prediction reveals Pbx3 and Meis1 as candidate regulators of neuronal regeneration

To better understand the factors governing NPCs fate decisions, our next goal was to identify transcription factors that could be directing fate transitions. We therefore integrated our ATAC-Seq data, scRNA-Seq data, and used bulk RNA-Seq data to predict gene regulatory networks occurring over regenerative time. We used HOMER to identify transcription factor (TF) binding sites in differentially accessible peaks associated with called GO terms (*Figure 3A–C*; *Heinz et al., 2010*). We then used our previously published bulk-RNA-Seq data (*Chang et al., 2017*) to confirm each HOMER-identified transcription factor and its targets were expressed at the timepoint of interest. Next we restricted our analysis to differentially accessible regions of the chromatin where TF binding sites were accompanied by differential gene expression in the single-cell RNA-Seq. Differential gene expression was performed between the uninjured and 24hpa timepoints in scRNA-Seq following parameters detailed in Materials and methods. These stringent criteria revealed two regulatory circuits spanning from 6hpa to 72hpa. One circuit predicts Meis1 as a regulatory factor that binds motifs near *runx2* and *etv1* at 24hpa, and Etv1 as a regulatory factor that binds motifs near *pbx3* at 24hpa (*Figure 6A*). Each of these four transcription factors also has numerous neuronal differentiation genes among its predicted targets. In our scRNA-Seq dataset, *etv1*, *meis1*, *pbx3*, and *runx2* are all expressed in the neural cells and are therefore reasonable candidates to regulate neural cell fate transitions. Of these, *pbx3* was the most restricted to the neural lineage (*Figure 4—figure supplement 1E,G*), and was found in several neuronal cell types. Pbx3 and Meis1 are homeodomain transcription factors thought to coregulate targets in B-cell leukemia and proximal-distal patterning of the regenerating axolotl limb (*Li et al., 2016*; *Mercader et al., 2005*). In *Xenopus*, Meis1 is well-studied for its roles in neural crest and posterior neural fate specification, working together with Pbx1 (*Kelly et al., 2006*; *Maeda et al., 2001*; *Maeda et al., 2002*; *Salzberg et al., 1999*). Neither *pbx3* nor *meis1* have been described in neural regeneration, and so we decided to probe the function of these genes in neural development and regeneration in *Xenopus*.

We assessed target regions and differential accessibility for Pbx3 and Meis1 in the ATAC-Seq data, and gene expression in the scRNA-Seq data. Considered across all regenerative timepoints, Pbx3 and Meis1 have both independent and overlapping targets. A large fraction (26/61) of Pbx3 targets are shared with Meis1 (*Figure 6B*). Meis1's target peaks are preferentially increased in accessibility at 24hpa, while Pbx3's are overall increased at 72hpa. Both transcription factors have numerous neuronal differentiation factors among their targets (*Figure 6C,D*). When we examined expression of *meis1* and *pbx3* in our scRNA-Seq data, we found that transcripts for both factors were detected in multiple differentiated neuronal types, initially at low levels and in a small proportion of cells (*Figure 6E*, and white portions of left pie charts in F). Following injury, at 24hpa, expression of *meis1* increased to 30% of all neural cells (*Figure 6F*, upper right), driven by increases in interneurons (IN), dopaminergic neuron (DN), and especially motor neuron (MN) clusters (*Figure 6E*, top). At 24hpa expression of *pbx3* increased to 11% of all neural cells (*Figure 6F* bottom right), driven by a strong increase in motor neuron expression (*Figure 6E*, bottom). Both factors are therefore expressed in cell types likely to be completing differentiation at 24hpa, most notably the motor neurons.

Previously, spatial expression patterns for *meis1* have only been reported up to stage 29 (*Maeda et al., 2002*) and expression patterns for *pbx3* have not been reported in *Xenopus*. Therefore, we used in situ hybridization to characterize their expression in the tadpole. At stage 41, both

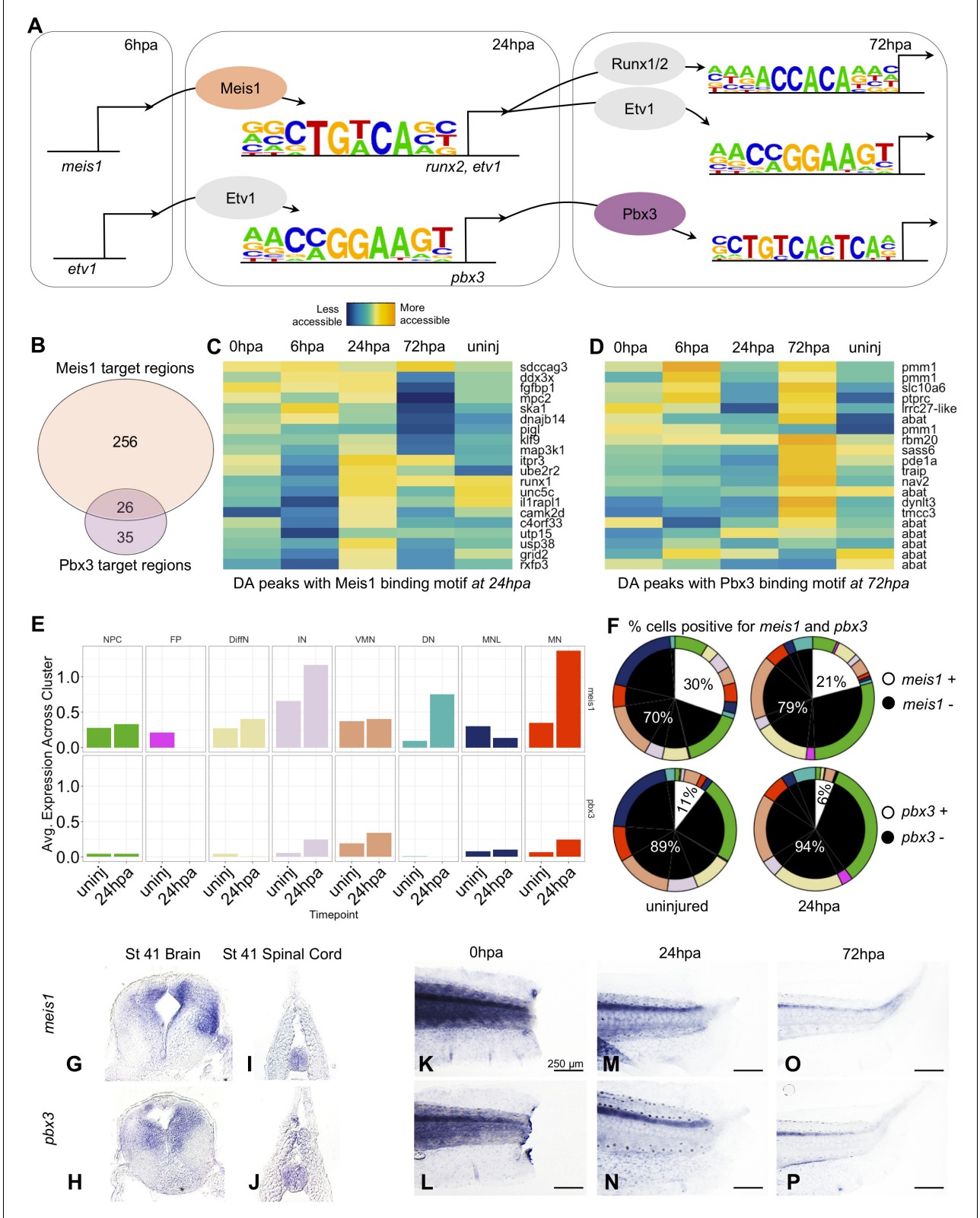

**Figure 6.** Gene regulatory network prediction identifies Meis1 and Pbx3 as key regulators of neural regeneration. (**A**) Predicted gene regulatory networks across regenerative time were derived from integrative analysis between ATAC-Seq and RNA-Seq data (See analysis in Materials and methods). Meis1 and Pbx3 emerged as candidate regulators of regeneration. (**B**) Venn diagram showing the number of Meis1 and Pbx3 target regions found in differentially accessible regions of the chromatin. The overlap represents regions where both Meis1 and Pbx3 binding motifs

*Figure 6 continued on next page*

*Figure 6 continued*

were found. (C, D) Heatmaps showing accessibility of 20 differentially accessible regions of the chromatin identified with binding sites for (C) Meis1 or (D) Pbx3. (E) Average gene expression of *meis1* and *pbx3* in each neural cell cluster in the scRNA-Seq data in the uninjured and 24hpa timepoints. (Abbreviations: Neural Progenitor Cell (NPC), Differentiating Neuron(DiffN), Interneuron (IN), Vulnerable Motor Neuron (VMN), Dopaminergic Neuron (DN), Motor Neuron Leptin+ (MNL), Motor Neuron (MN)). (F) Sunburst diagram showing the percentage of neural cells that are either positive or negative for *meis1* or *pbx3* gene expression (inside) and the percentage of each neural cluster positive or negative for the genes (outside). (G–P) In situ hybridizations for *meis1* and *pbx3* in transverse sections of the head (G, H) and tail (I, J) at stage 41. In situ hybridizations for *meis1* and *pbx3* at (K, L) 0hpa, (M, N) 24hpa, and (O, P) 72hpa.

*meis1* and *pbx3* are expressed in the uninjured brain and spinal cord (*Figure 6G–J*). In the posterior tail, expression of both factors is weak and diffuse immediately following injury (*Figure 6K,L*). However, by 24hpa, expression of both factors increases in the spinal cord (*Figure 6M,N*). By 72hpa, expression persists more weakly in the spinal cord. These analyses therefore indicate that *pbx3* and *meis1* are expressed in the spinal cord, increase in neuronal subtypes at 24hpa, and contribute to increased promoter accessibility at their target genes at 24hpa (particularly for Meis1) and 72hpa (particularly for Pbx3), some of which are likely co-regulated by both factors.

## Meis1 and Pbx3 are necessary for successful spinal cord and tail regeneration

We next asked whether Meis1 and Pbx3 are independently required for regeneration. To do so, we first injected morpholinos blocking translation of either *meis1* or *pbx3* into the dorsally fated blastomeres of 4 cell stage embryos to knockdown expression in the neural lineage (*Figure 7—figure supplement 1B*). Knockdown of either Meis1 or Pbx3 resulted in a similar phenotype in stage 41 tadpoles that included small or missing eyes, a shorter anteroposterior body axis and reduced pigment cell population (*Figure 7—figure supplement 1C,D*). We followed up on these phenotypes by repeating the injections with a second set of morpholinos against Pbx3 and Meis1. The second set of morpholinos phenocopy the original morpholino injections (*Figure 7—figure supplement 1E,F*). As a further confirmation, we designed and injected CRISPR guide RNAs targeting each of these genes together with Cas9 protein at the 4 cell stage, and were able to phenocopy the morpholino effects with this strategy as well (*Figure 7—figure supplement 1G–J*). When we stained stage 41 morphants with anti-neurofilament antibody, we found that their spinal cord and intersomitic axons were mispatterned, suggesting Meis1 and Pbx3 are necessary for proper development of the nervous system (*Figure 7—figure supplement 2B–F*). To bypass the early embryonic effects of Meis1 and Pbx3 knockdown, we designed the same MO sequences as tissue-permeable vivo-MOs. We injected either Pbx3 or Meis1 vivo-MO into the embryonic tail vein at the late tailbud stage (NF 35), and reared these embryos to stage 41 (*Figure 7A*). Neither vivo-MO caused gross morphological defects when delivered this way, but both vivo-MOs resulted in reduced axons in the spinal cord as assayed by neurofilament stains at stage 41 (*Figure 7C,D*), relative to control tadpoles injected with tracer only (*Figure 7B*).

To specifically assay the effect of Meis1 or Pbx3 knockdown on regeneration, we injected either Meis1 or Pbx3 vivo-MOs at stage 35 as described above, reared these embryos to stage 41 and amputated tails. At 72hpa, both Pbx3 and Meis1 knockdown tadpoles have reduced and disorganized neurofilament staining relative to tracer-injected controls (*Figure 7F,G*, compare with E). We quantified the length of the regenerated tail in control and knockdown tadpoles at 72hpa, and found that both Meis1 and Pbx3 knockdown tadpoles have significantly shorter regenerated tails and spinal cords relative to tracer-injected control tadpoles (*Figure 7K*). Spinal cord area (measured in confocal optical longitudinal sections) is also significantly reduced in both morphants at 72hpa (*Figure 7L*), although the density of PH3 positive nuclei is not significantly changed relative to controls (*Figure 7H–J,M*). Embryos injected with either Meis1 or Pbx3 MOs at the 4 cell stage also caused neurofilament defects during regeneration, as well as shortened regenerated tails and spinal cords (*Figure 7—figure supplement 2G*-AA), although, we cannot rule out that some aspects of these phenotypes may be attributable to embryonic growth and patterning defects in the starting tail tissue. We conclude that Meis1 and Pbx3 are not only required for proper neural development but also for proper regeneration of the tail and spinal cord.

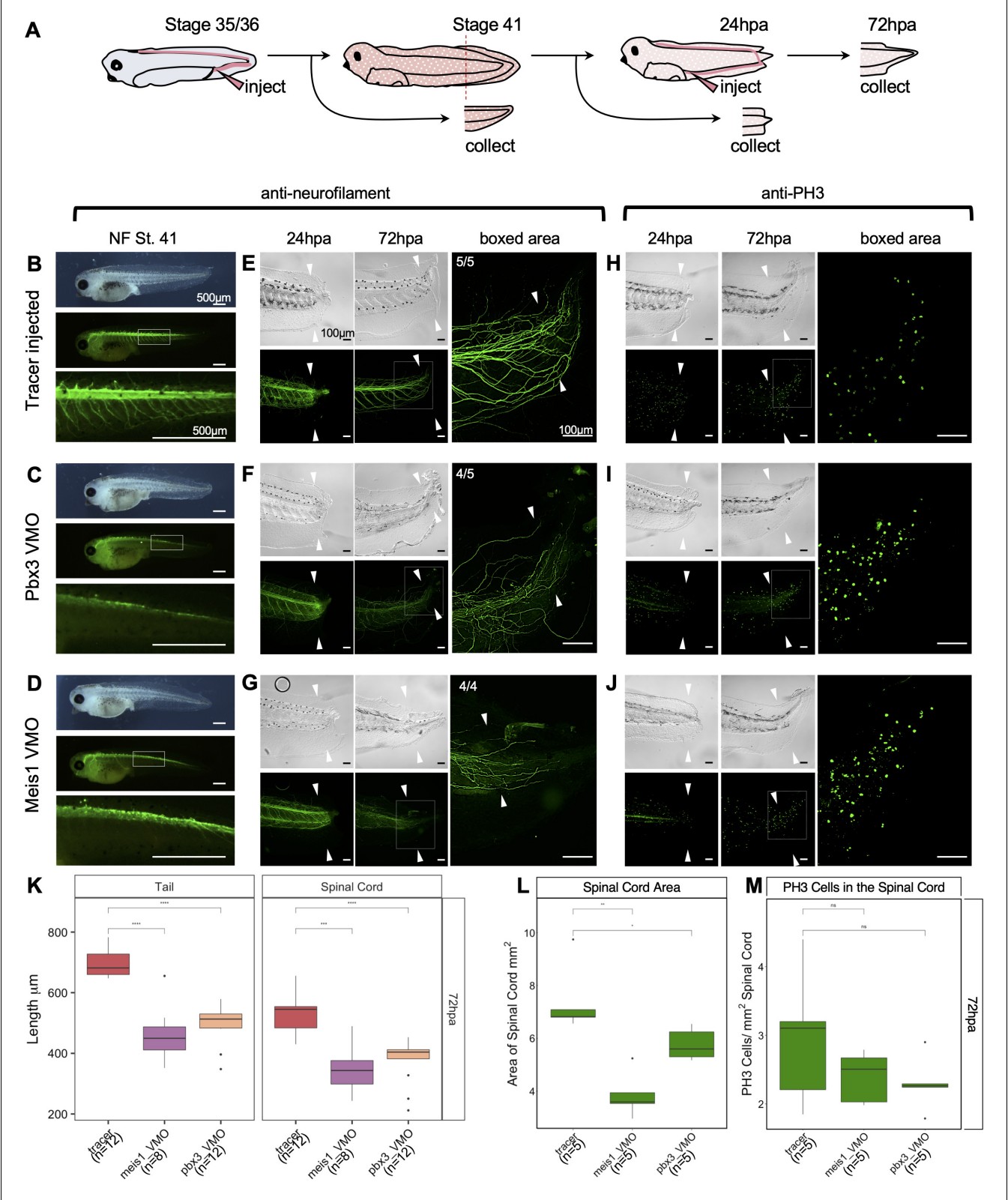

**Figure 7.** Pbx3 and Meis1 are independently required for successful regeneration of neural tissues and tails in response to injury. (**A**) Injection scheme for administering vivo-morpholino. Stage 35 tadpoles were injected with a tracer or vivo-morpholinos (VMO) targeting *meis1* or *pbx3* and allowed to grow 24 hr to stage 41. Stage 41 tadpoles were amputated and 24hpa regenerates were collected. (**B–D**) Stage 41, whole-mount tadpoles shown in brightfield and immunostained against neurofilament. The box in the middle image corresponds to the enlarged image below. These images are

*Figure 7 continued on next page*

*Figure 7 continued*

shown for (**B**) tracer injected, (**C**) Meis1 VMO injected, and (**D**) Pbx3 VMO injected tadpoles. (**E–J**) 24hpa and 72hpa regenerates were collected and stained for neurofilament (**E–G**) or PH3 (**H–J**). Top photos are in DIC and bottom photos are immunostained as indicated. White arrows indicate amputation plane. These images were collected for (**E, H**) tracer, (**F, I**) Meis1 VMO, and (**G, J**) Pbx3 VMO. (**K**) Regenerated tail and spinal cord lengths were measured and reported in boxplots. (**L**) Boxplot representing regenerated spinal cord area. (**M**) Boxplot representing PH3 cells per regenerated spinal cord area. Statistics represent a two-tailed t-test to determine significance between conditions. (ns = not significant, *<0.05, **<0.005, ***<0.0005, ****<0.00005).

The online version of this article includes the following source data and figure supplement(s) for figure 7:

**Source data 1.** Regenerated Tail Length Data for Vivo-Morphants.
**Source data 2.** PH3 Cell Count and Regnerated Spinal Cord Area for Vivo-Morphants.
**Figure supplement 1.** Embryonic Meis1 and Pbx3 morphants are characterized by small heads and missing or small eyes.
**Figure supplement 2.** Meis1 and Pbx3 are independently necessary for successful regeneration of the axial tissue and neuronal patterning.
**Figure supplement 2—source data 1.** Regenerated Tail Length Data for Embryonic Morphants.

# Discussion

## Neural lineage-specific analysis of regeneration identifies new regulatory factors and target genes

The full regeneration of a lost structure requires integrated gene regulatory decisions by multiple cell types. These have been documented through genome-wide transcriptomic analyses of the entire regenerating structure, such as the *Xenopus* tail (*Chang et al., 2017*; *Love et al., 2011*), spinal cord (*Lee-Liu et al., 2014*), or axolotl limb (*Dwaraka et al., 2019*). However, in bulk analyses it remains uncertain which cell types are up- or down-regulating particular genes, and increased expression in one cell type may be canceled out by decreases in another. These concerns are equally problematic in bulk-tissue analysis of gene regulatory dynamics (through methods such as ChIP-Seq or ATAC-Seq). In some cases, this problem can be addressed by explanting the tissue of interest (*Chung et al., 2014*), but for rare cell types or morphologically complex structures this is not a tenable solution. Here we present a major advance for vertebrate regeneration by using flow-cytometry and scRNA-Seq to coordinately study chromatin accessibility and gene expression in neural progenitors during regeneration, a critical cell type for spinal cord regeneration (*Fei et al., 2014*; *Gaete et al., 2012*; *Muñoz et al., 2015*). By isolating neural progenitors, we are able to identify a unique chromatin landscape for these cells, including regulatory peaks that cannot be detected in bulk tissue analysis.

It is important to note the potential limitations of our analysis as well. While our FACS approach utilizing the *pax6:GFP* transgenic line captures a broad domain of neural progenitor cells, we recognize that not every NPC may have been captured. For example, there may be some Sox2+ NPCs that are *pax6*-, and these cells may contribute to regeneration in a distinct way that we did not capture. In particular, our confocal sections (*Figure 1*) suggest that the *pax6:GFP*+ domain captures a large window of progenitors along the dorsoventral axis but does not capture the dorsal most Sox2 + NPCs. Future analyses using other NPC reporters, as well as domain-specific reporters for region-specific subtypes of both neural progenitors and differentiated neurons, would be necessary to fully round out the picture of what transcriptional priorities are shared among progenitor subtypes and which are region-specific. We may have also captured cells that were *pax6*+ and Sox2-. Specifically, we may have captured some cells that were recently *pax6*+ neural progenitors, and are now differentiating but retain GFP protein. This hypothesis would be well informed by birthdating neurons in the transgenic line to visualize if newborn neurons retain reporter GFP; however, for this set of experiments we acknowledge this potential caveat. Finally, while we find that Sox2 and *pax6:GFP* colocalize closely over much of their respective domains at stage 41, the very small size and delicacy of the regeneration bud at 24hpa precluded verification that these two markers continue to colocalize in the same way during the early stages of regeneration as the neural ampulla forms. Therefore while we expect that the spatial domains of these markers continue to overlap, and our scRNA-SEQ confirms that NPCs continue to express both of these markers, there may be subtleties to their spatial dynamics that reveal principles of NPC organization in regeneration that we did not capture.

Our study identified both novel candidate markers of specific neural lineages, and potential regulatory circuits directing regenerative decisions. Many of the genes we identified as differentially

expressed and with differentially accessible promoters were not previously detected in bulk RNA-Seq or microarray analyses (*Chang et al., 2017*; *Love et al., 2011*). Of these, *pbx3* stands out as a gene with expression that is well-restricted to the neural lineage, becomes further restricted to differentiated neurons following injury, has a differentially accessible promoter itself, and has a target motif upstream of multiple differentially-expressed neuronal genes. Nevertheless, *pbx3* was not detected as differentially expressed in previous analyses of regeneration, likely because of the low overall expression levels of *pbx3*, and because expression changes are lineage-specific. Only by combining motif prediction from ATAC-Seq and the high-resolution expression data from scRNA-Seq did we bring this transcription factor to the top of our candidate list. Similarly, *meis1*, which is well-known as a regulator of neural and neural crest development (*Erickson et al., 2010*; *Kelly et al., 2006*; *Machon et al., 2015*; *Mojsin and Stevanovic, 2010*; *Rataj-Baniowska et al., 2015*; *Stedman et al., 2009*; *Yamada et al., 2013*), has not previously been detected in differential expression analyses during *Xenopus* regeneration. This may be because *meis1* is more broadly expressed, and bulk RNA-Seq was unable to capture the heterogeneity of expression changes *meis1* undergoes across multiple cell types.

## Evidence for a temporal uncoupling of differentiation and proliferation in the regenerating neural lineage

We find that *pax6+* neural progenitors place an early priority on neuronal differentiation, and specifically on formation of interneurons and motor neuron subtypes by 24hpa, shown as a model in *Figure 8A–C*. This is supported by the widespread accessibility enrichment of neuronal differentiation genes specifically at 24hpa, by the decline in relative proportion of neural progenitors and transitioning neurons in favor of motor neurons and interneurons, and by a reduction in the number of cells in S/G2/M phases of the cell cycle.

Our finding that *pax6+* progenitors prioritize differentiation before proliferation in regeneration represents an intriguing counterpoint to embryonic development, in which regional-specific proliferative neural progenitors can only later make the decision to undergo neurogenesis and exit the cell cycle (*Hardwick et al., 2015*). Transcriptional analysis of proliferating blastemal cells as well as recent pre-publication work in later-stage tadpoles of *Xenopus laevis* suggests the same model may also hold true in that species (*Pelzer et al., 2020*; *Tsujioka et al., 2015*).

The cell cycle exit and differentiation of some of the existing pool of neural progenitors and differentiating neurons at 24hpa results in a decline in these cell types and an increase in motor and interneurons. Not all progenitor cells exit the cell cycle however, and a residual population remains, from which we propose that additional progenitors are made by proliferation after 24hpa, in agreement with the emphasis placed on progenitor proliferation seen by ATAC-Seq at 72hpa and mitotic analysis by PH3 (*Figure 8D*). We expect that the organized formation and outgrowth of neuronal structures remains important from 24 to 72hpa, as axons visibly elongate, and that *pbx3* and *meis1* are important regulators of these processes based on their target gene accessibility profiles (*Figure 8E/F*). Our model agrees with numerous findings that there is little proliferation after tail amputation prior to 48hpa (*Contreras et al., 2009*; *Gaete et al., 2012*; *Love et al., 2011*; *Pelzer et al., 2020*; *Tsujioka et al., 2015*) and with bulk RNA-Seq analysis showing a transcriptional upregulation of axonogenesis and other neuronal differentiation genes (*Chang et al., 2017*). Our analysis suggests that early changes in neural cell identity and distribution, as well as the genes driving them, may represent an important early phase of regeneration that precede proliferation. In other regenerative vertebrates, specifically the anole lizards and geckos, spinal cord regeneration is incomplete; it includes formation of new axons along an ependymal cell tube, but does not include proliferation of progenitor cells to make a complete spinal cord (*Duffy et al., 1990*; *Tokuyama et al., 2018*; *Wang et al., 2012*). An interesting future direction would be to pursue whether program supporting axonogenesis may therefore have persisted in amniote lineages, where other features of complete spinal cord regeneration have been lost.

## Neural cell types preserve their identities in the regenerating *Xenopus* tail

The identities of specific neural cell types are readily assigned to the same clusters both before and after tail transection in *Xenopus*. This agrees with recent single-cell transcriptomic analysis of the

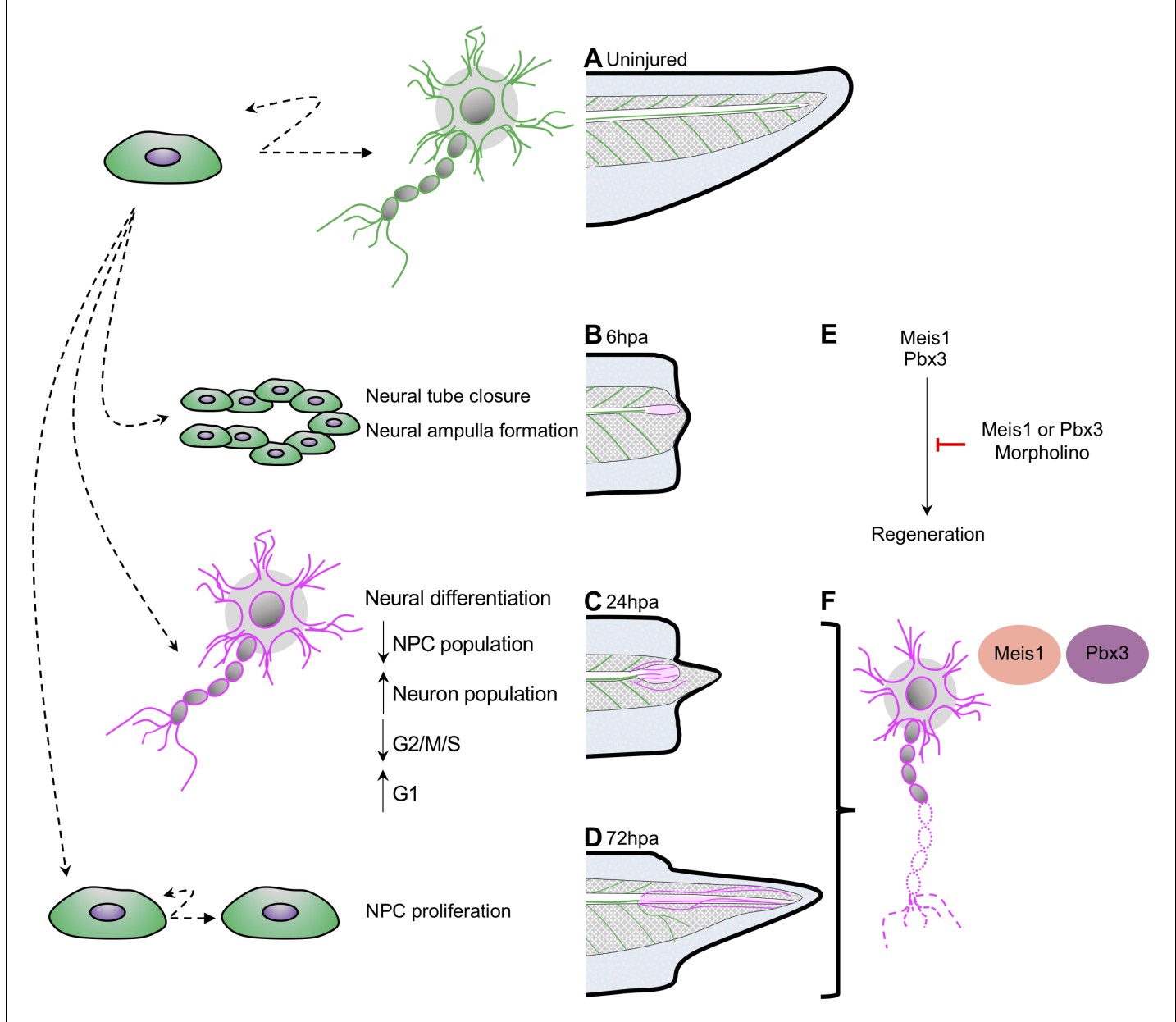

**Figure 8.** Model for neural progenitor regeneration. (**A**) In normal development, neural progenitor cells make cell fate decisions to proliferate or differentiate. From this study we predict from our ATAC-Seq and scRNA-Seq analysis that neural progenitors first prioritize (**B**) migration and tube morphogenesis at 6hpa, followed by (**C**) neural differentiation at 24hpa, and then (**D**) turn on proliferation programs at 72hpa. (**E**) From integrative analysis of these data sets, we identified *meis1* and *pbx3* as candidate regulators of NPC regeneration at 24hpa and 72hpa. Loss-of-function experiments suggest Meis1 and Pbx3 are necessary for successful regeneration as well as proper axonal patterning. Magenta shaded spinal cord represents NPCs in regenerated spinal cord. Magenta axons represent axons in the regenerated spinal cord and tail tissue.

regenerating *Xenopus laevis* tail and spinal cord, in which the majority of cell types can be clearly assigned at 1, 2 and 3 days after amputation (*Aztekin et al., 2019*; *Pelzer et al., 2020*). The predominant model for vertebrate regeneration remains the axolotl limb, in which the formation of a morphologically undifferentiated mesenchymal blastema has been well characterized through morphological, histological, transcriptional, and single-cell analysis (*Currie et al., 2016*; *Gerber et al., 2018*; *McCusker et al., 2015*). In the axolotl limb, scRNA-Seq shows that different connective tissue cell types lose their unique transcriptional identities early in regeneration and converge on a similar signature, as the blastema forms (*Gerber et al., 2018*). While the early *Xenopus* tail regenerate has

also been described as a blastema (*Gargioli and Slack, 2004*; *Tsujioka et al., 2015*), its parallels to the axolotl blastema have been debated (*Mochii et al., 2007*), and our study confirms that diverse cell types retain well-discriminated transcriptional signatures as regeneration proceeds. This suggests that the molecular nature of the blastema may differ substantially between axolotls and *Xenopus*. In other regenerative contexts such as bone regeneration of the zebrafish fin, the formation of a blastema at the tip of each bony ray has been well documented (*Knopf et al., 2011*; *Münch et al., 2013*; *Sousa et al., 2011*). Global transcriptional responses to injury can therefore vary among regenerative vertebrate structures, which highlights the importance of multiple models for appendage regeneration.

## Pbx3 and Meis1 are required for axon organization and for tail regeneration

We identify two transcription factors, Pbx3 and Meis1, as transcriptional regulators acting upstream of neuronal differentiation genes during regeneration (*Figure 8E*). We show that these factors are both expressed in the regenerating spinal cord, with increased expression in motor neuron subtypes during regeneration. Loss of function of either factor results in disorganized neurofilament outgrowth with a failure of successful axial tissue regeneration. These factors are predicted to bind upstream of several dozen differentially accessible neuronal differentiation factors and are thus well-placed to drive the prioritization of neuronal differentiation we observe. Loss of either factor does not result in a failure to form neurofilaments, but does result in grossly disrupted organization of neurofilaments and reduced regeneration of the tail and spinal cord. Our data therefore suggests these factors may be critical for the proper re-formation and organization of differentiated neuronal structures. We note that even though the differentially accessible targets of Pbx3 are preferentially found at 72hpa, when proliferation is increasing, many targets of Pbx3 are shared with Meis1, and Pbx3 itself increases in expression in neuronal types by 24hpa. We therefore propose that Pbx3 and Meis1 contribute to the ongoing differentiation of neuronal structures from 24hpa to 72hpa (*Figure 8F*). Intriguingly, loss of either Pbx3 or Meis1 results in reduced tail regeneration overall. This may be due to interactions between regenerating nerves or neuronal structures and other axial structures, as regeneration of appendages has been shown to be nerve dependent in *Xenopus* and other systems (*Farkas and Monaghan, 2017*).

Pbx3 and Meis1 may work independently or as part of a complex. Pbx3 and Meis1 are known to heterodimerize and coregulate targets upstream of *hoxa9* during leukemia progression (*Li et al., 2016*; *Machon et al., 2015*). Meis1 also partners with the related protein Pbx1 in axolotl limb regeneration and neural development (*French et al., 2007*; *Mercader et al., 2005*; *Mojsin and Stevanovic, 2010*). Although their targets are incompletely overlapping in our regeneration data, they are co-expressed in several cell types, and so we do not rule out the possibility that they may also act as a complex in the tadpole tail. Neither *pbx3* nor *meis1* have been studied in neural regeneration. These two factors therefore represent exciting new candidates for regulating neuronal differentiation in regeneration.

## Conclusion

Taken as a whole, our study presents a coordinated analysis of chromatin accessibility and single cell transcriptomics in the regenerating neural lineage. We show that neural progenitors temporally uncouple their regulation of differentiation and proliferation, placing an earlier priority on differentiation of motor neuron subtypes. This new model represents an intriguing contrast to embryonic development. We computationally identify multiple candidate regulators of neural progenitor fate decisions in a regenerative context, and specifically highlight *pbx3* and *meis1* as critical regulators of neural development and regeneration.

## Materials and methods

Key resources can be found in the key resource table, *Supplementary file 2*. All data generated and analyzed in this paper can be found with the GEO accession GSE146837 (https://www.ncbi.nlm.nih.gov/geo/query/acc.cgi?acc=GSE146837). Source files and analysis code can be found at https://gitlab.com/akakebee/kakebeen-et-al-2020 (*Kakebeen, 2020*; copy archived at https://github.com/elifesciences-publications/kakebeen-et-al-2020).

## X. tropicalis husbandry and use

Use of *Xenopus tropicalis* was carried out under the approval and oversight of the IACUC committee at UW, an AALAC-accredited institution. Ovulation of adult *X. tropicalis* and generation of embryos by natural matings were performed according to published methods (*Khokha et al., 2002*; *Sive et al., 2010*). Fertilized eggs were de-jellied in 3% cysteine in 1/9x modified frog ringer's solution (MR) for 10–15 min. Embryos were reared as described (*Khokha et al., 2002*. Staging was assessed by the *Nieuwkoop and Faber (1994)* staging series. In this study we used both wild-type frogs and frogs from the triple transgenic line Xtr.Tg(pax6:*GFP*;cryga:*RFP*;actc1:*RFP*)Papal (*Hartley et al., 2001*; *Hirsch et al., 2002*; RRID:NXR_1.0021) reared and purchased from the National *Xenopus* Resource (https://www.mbl.edu/xenopus/; RRID:SCR_013731). *Pax6* transgenic matings were performed by crossing a heterozygous transgenic frog to a wild-type frog. These matings yielded clutches with 50/50 wt/*pax6:GFP*+ populations.

## Sectioning and imaging

For sectioning, tadpoles were fixed for 30 min in 10xMEMFA, 3.7% formaldehyde at room temperature. Tadpoles were then washed briefly in PBT and incubated in 4% sucrose in 1xPBS at 4C overnight. Tadpoles were embedded in Tissue Tek O.C.T. (Sakura 4583) in 10mmx10mmx5mm molds (Sakura 4565) and stored in the −80C until sectioning. Tadpoles were sectioned transversely at 14 µm with a cryostat (Leica). Sections that were not stained were baked onto slides on a heatblock and imaged within 24 hr to capture endogenous *pax6* reporter fluorescence. Images were acquired with a Lecia SP8 and UPDATE camera.

## X. tropicalis amputation assay

*X. tropicalis* were sorted for transgenic reporter fluorescence for regeneration assays. Tadpoles were anesthetized with 0.05% ms-222 in 1/9x MR and tested for response to touching prior to amputation surgery. Once fully anesthetized, a sterilized scalpel was used to amputate the posterior third of the tail. Amputated tadpoles were removed from anesthetic media within 10 min of amputation into new 1/9x MR. Tadpoles were left to regenerate at a density of no more than 3 tadpoles to 1 mL of media.

## Cell dissociation and fluorescent activated cell sorting (FACS)

### Cell dissociation

At each timepoint, regenerating tadpoles were anesthetized until non-responsive to touch. A scalpel was used to amputate to the regenerated tail tissue (most posterior point to ~500 µm anterior of that point). Regenerated tail tips were collected and spun down at 1000xg. Media was taken off and ~60 tails were resuspended in 200 uL of 0.035 mg/ml liberase in PBS (Roche 05401119001). Tails were left to incubate for 20 min at room temperature. A p-200 was then used to pipet the tails up and down gently and break up tail tissue. Once no visible chunks were apparent, cells were spun down for 3 min at 1000xg. Supernatant was disposed and cells were washed with 180 uL of 1xPBS. Cells were then spun once more for 1 min at 1000xg and supernatant disposed. Cells were resuspended in 200–300 uL of 1xPBS and filtered through a 70 µm cell filter (Fisher 22-363-548) into a FACS tube. Cells were kept on ice until sorting.

### FACS

Cell sorting and collecting were performed through on the BD FACS Aria III Cell Sorter (BD Biosciences). Wild-type samples were used as negative controls to set sorting gates to exclude any autofluorescence from our true GFP signal. Cells were sorted based on selection from three criteria. FSC x SSC were used to select for cells based on size and granularity. From this population, cells were visualized with the FSC-W x FSC-H dimensions to select for singlets and exclude doublets. The doublets were then visualized in a histogram where GFP was the x-axis. For GFP gate setting, selection gates were drawn for all area without cells in the wild-type tails. Here we assume that any GFP-signal is autofluorescence and need be selected against. Positive control transgenic tadpoles were used to test for GFP signal. Cells were passed through the sorter and collected through a 70 µm nozzle. Cells were collected into 50 uL of 1xPBS. For *pax6* samples, GFP+ and GFP- cells were collected

into separate tubes and for all-tissue samples, GFP+ and GFP- cells were collected into the same tube.

## Quantitative RT-PCR

For RT-PCR, either GFP+ cells or GFP- cells were collected from the flow cytometer directly into RNA lysis buffer. Between 2000–4000 cells were collected for each condition from uninjured tail tips. Cells were then agitated at 42C for 45 min. RNA/DNA were extracted and purified from the lysed cells by phenol:chloroform extraction with subsequent ethanol precipitation. Samples were then DNAse treated to remove DNA. To make cDNA from the RNA, we used the Superscript III reverse transcriptase kit (ThermoFisher 18080044). Amplifications were carried with iQ SYBR Green Super-mix (BioRad 1708882) for 40 cycles of: 95C:10 s, 57C:30 s after initial 2minutest at 95C. Primers used for *gfp, pax6, actc1, tubb2* are provided *Supplementary file 2*.

## ATAC-Seq library preparation

Within 30 min of cell sorting, cells were prepared for transposition. (See *Supplementary file 1* for details of cells sorted for each sample). Cells were spun down for 3 min at 1000xg and supernatant was discarded. Cells were resuspended in 15 uL of TN5 reaction mix using the TN5 enzyme, buffer, and water from the Nextera DNA Sample prep kit (Illumina 15028212). Cells were transposed for 1 hr at 37C. Post transposition, cells were removed from 37C and DNA was purified using the Qiagen MinElute kit (Qiagen 28206). DNA was eluted in 10 uL of buffer EB. Purified DNA was then amplified using the 2xNEB PCR Master Mix (NEB M0541L), a universal forward primer, and an indexed reverse primer from the i7 Illumina series. Initial amplification as follows: 72C:5 min, 98C:30 s, then 5 cycles of 98C:10 s, 63C:30 s, 72C:1 min. 5 uL of the initial PCR product was then used for a 15 µl side qPCR reaction to determine how many more cycles to run to stop amplification prior to saturation. The side reaction was carried out as follows: 98C:30 s, then 30 cycles of 98C:10 s, 63C:30 s, 72C:1 min. The number of cycles was determined by taking the range of the starting and ending CT values and dividing that number by 4, then identifying how many cycles the sample had gone through at the derived CT value. The initial PCR reactions were then run for the calculated number of extra cycles and removed from the thermo cycler. Amplified samples were purified with the Qiagen MinElute kit and eluted in 20 uL of buffer EB. 5 uL of purified PCR product were run out on a 5% acrylam-ide gel in 1xTBE at 100V for ~45 min. Gels were stained with Ethidium Bromide and imaged on the gel imager (BioRad). Samples with sufficient periodic bands, relating to length of DNA to wrap a mono-, di-, tri- nucleosome, were then run out on a bioanalyzer. Library concentrations were taken using the Qubit high sensitivity dsDNA assay. Libraries were pooled by normalizing library concentrations to the lowest concentration and adding equal volumes of libraries for a minimum of 5 ng of each library.

Libraries were sequenced on the Illumina Next-Seq platform across three NextSeq 500 High Output Kit v2.5 (150 cycles, Illumina CAT. 20024907). Libraries were sequenced with, paired end, 75 bp reads at the Sound Genomics facility.

## ATAC-Seq analysis pipeline

### Trimming adaptors and alignment

Adapters were trimmed from reads and low-quality sequences (Phred < 33) were removed using Trim Galore!. Reads were aligned to xtropicalis9.0 using Bowtie2 (option:–very-sensitive) (*Langmead and Salzberg, 2012*; *Langmead et al., 2019*). Duplicate reads were marked using Picard 'MarkDuplicates' (http://broadinstitute.github.io/picard/). Duplicate reads were removed using SAMtools. Bigwigs were generated using a custom script (https:// rpubs.com/achitsaz/98857) and visualized in IGV.

### Peak calling

Peaks were called using MACS2 (options:–nomodel–shif −100–extsize 200) (*Zhang et al., 2008*). Consensus peak set used for differential accessibility were called on a merged bam of all ATAC-seq files used in this experiment.

## Peak annotation

Annotation and genomic feature enrichment analysis. Annotation of ATAC-seq peaks and genomic annotation enrichment analysis were performed using GenomicRanges (*Lawrence et al., 2013*), matching each peak to the nearest TSS using Xtropicalisv9.0.Named.primaryTrs.gff3 from Xenbase.

## Differential accessibility analysis

Differential accessibility analysis in ATAC-seq peaks. Differential accessibility across ATAC-seq sample groups was determined as detailed in the edgeR users guide (*Robinson et al., 2010*). To define the regions for differential accessibility analysis, peaks were called on a merged bam of all ATAC-seq samples in this experiment. A matrix of counts for all samples was then generated using the 'featureCounts' in the Rsubread package (options: isPairedEnd = TRUE, maxFragLength = 2000). The counts matrix was filtered to select rows having at least one count per million in n−one samples to minimize the influence of variability at the threshold for sensitivity on the analysis.

## Gene Ontology analysis

### GO:BP analysis (gprofiler2)

Using differential accessibility flags, we identified all regions more accessible in the indicated contrast. The TSS annotation of these regions was used as input for gene ontology analysis. GO analysis was carried out using the 'gprofiler2' R package (*Raudvere et al., 2019*). The command 'gost' was used to call GO terms (options: query = DA peak list, organism="hsapiens', sources="GO:BP', evcodes = TRUE).

### Reduce Redudancy (ReviGO)

To reduce redundancy of GO terms, we used the online resource ReviGO (http://revigo.irb.hr/) (*Supek et al., 2011*). We input GO IDs and p-values into ReviGO and used the following parameters to run the analysis: allowed similarity = tiny(0.4), p-values, database with GO term sizes = *Homo sapiens*, Semantic similarity measure = SimRel. We then saved the output. CSV file for further plotting and analysis in R.

## Single cell RNA-Seq library preparation

GFP+ cells from ~500 tails were collected for single cell analysis using the above methods but scaled for 500 tails. After sorting, cells were immediately washed with 1xPBS and diluted to a concentration of 1000 cells/µl based on the 10x Genomics guidelines. We aimed for capture of 10,000 cells using the 10x Genomics Platform. Single-cell mRNA libraries were prepared using the single-cell 3' solution V2 kit (10x Genomics). Quality control and quantification assays were performed using a Qubit fluorometer (Thermo Fisher) and a D1000 Screentape Assay (Agilent). Libraries were sequences on an Illumina NextSeq 500 using 75-cycle, high output kits (reads 1: 26 cycles, i7 Index: eight cycles, read 2: 57 cycles). Each sample was sequenced to average read depth of 16 million total reads. This resulted in an average read depth of ~15,000 reads/cell after read depth normalization.

## Single cell RNA-Seq data processing

Cellular barcodes and µMis were determined using Cell Ranger 2.0.2 (10X Genomics) and cells were filtered to include only high-quality cells. Cell ranger defaults for selecting cell-associated barcodes versus barcodes associated with empty partitions were used.

## UMAP visualization and clustering

### Clustering and integration of all cells in uninjured tail and 24hpa

We used Seurat V3.0 for analysis of scRNA-Seq data (*Butler et al., 2018*; *Butler et al., 2018*). To reduce dimensions and cluster cells for the uninjured and 24hpa data sets, we followed Seurat's 'Integrating stimulated vs. control PBMC datasets to learn cell-type specific responses' vignette (https://satijalab.org/seurat/v3.1/immune_alignment.html). We created Seurat objects from Cell-Ranger processed data with the command 'CreateSeuratObject' (options: min.cells = 5). We then removed cells that had less than 500 RNA features by subsetting data (options: subset = nFeature_RNA >500). Cells were normalized ('NormalizeData') and variable features were

found with 'FindVariableFeatures' (options: selection.method = 'vst'). Integration anchors for the two data sets were identified with 'FindIntegrationAnchors' (options: dims = 1:20) and the integration of the data sets was performed with 'IntegrateData' (options: dims = 1:20). The integrated data set was then scaled ('ScaleData') and principle components were found with the command 'RunPCA' (options: npcs = 60). We then reduced dimensions using Uniform Manifold Approximation and Projection (UMAP). We did so by using 'RunUMAP' (options: npcs = 40). Next we clustered cells by first finding nearest neighbors with 'FindNeighbors' (options: reduction="pca', dims = 40) and then calling clusters with 'FindClusters' (options: resolution = 0.7).

## Subsetting of neural cells in uninjured tail and 24hpa

We used neural marker genes (*sox2, neurog1, neurod1,* and *elavl4*) to determine which cell clusters were neural cells. To subset the data set, we used 'SubsetData' (options: ident.use = c(1,4,5,13,15). The subset data was scaled, and principle components were called. 'RunUMAP' and 'FindNeighbors' was run (options: reduction="pca', dims = 1:20). 'FindClusters' was then run to establish neural clusters (options: resolution = 0.4).

## Marker identification and differential expression analysis

### Cluster markers

To identify genes that define each cluster, we used Seurat's 'FindMarkers' command (options: ident.1 = Cluster of interest, min.pct = 0.3, only.pos = TRUE, assay = 'RNA'). This command was run for all seven subset neural clusters.

### Differential gene expression across timepoints

To identify genes that were differentially expressed between uninjured and 24hpa tail cells, the Seurat function 'FindMarkers' was used. (options: ident.1 = Cluster of interest_uninjured, ident.2 = Cluster of interest_24hpa, min.pct = 0.3, assay = 'RNA').

## Cell cycle prediction analysis

### Cell cycle prediction

To predict the cell cycle of cells sequenced in our scRNA-Seq, we followed the methods outlined in the Seurat Vignette 'Cell-Cycle Scoring and Regression' (https://satijalab.org/seurat/v3.1/cell_cycle_vignette.html). Genes defining 'S' phase and 'G2M' phase were called from a pre-loaded list of cell cycle markers from *Kowalczyk et al. (2015)*. These genes were then used in the Seurat command 'CellCycleScoring' to predict cell cycle phases (options: s.features = s.genes (from Seurat), g2m.features=g2 m.genes(from Seurat), set.ident = TRUE).

## Gene regulatory predictions

### Motif Finding

We used the 'findMotifsGenome.pl' command to identify all known binding motifs from the Homer motif library in differentially accessible peaks (*Heinz et al., 2010*). The motifs were then appended to the ATAC-Seq differential accessibility table by matching peak names. This table gave us a called peak, the TSS annotation of that peak, motifs called in the peak, and transcription factor binders of the called motif.

### Bulk RNA-Seq integration

TSS annotations were used to vet table for genes that are expressed at the given timepoint. To do so, we subsetted the table for genes that were expressed in bulk RNA-Seq data from *Chang et al. (2017)*.

### GRN prediction across regenerative time

Using a homemade script, we identified DA regions at 6hpa that had binding motifs in 24hpa DA peaks. We then identified DA regions at 24hpa that had binding motifs in 72hpa DA peaks. To connect the timepoints, we identified targets of TFs with motifs at 24hpa, then asked whether those targets had binding motifs at 72hpa. To narrow down potential GRNs to follow up on, we asked

whether the genes were expressed in neural cells in the scRNA-Seq data in the uninjured at 24hpa cells and whether they were differentially expressed.

## Whole-Mount in situ hybridization

Embryos and larval tadpoles were fixed in 1xMEMFA, 3.7% Formaldehyde for 2–6 hr at room temperature or overnight at 4C. *X. tropicalis* multibasket in situ hybridization protocols were followed as described in *Khokha et al. (2002)*. For plasmids used for in situ hybridization probes, see reagents table). Whole mount in situ tadpoles were imaged on a Leica M205-FA with a D5C550 camera.

## Morpholino microinjections

Morpholinos were ordered from Gene Tools. BLAST searches for both morpholinos for Meis1 and Pbx3 using the *X. tropicalis* v9.1 genome identified just one target in the correct gene for each of the 4 MOs, and no off target binding elsewhere in the genome.

### Dorsal blastomere injections

Wild-type *X. tropicalis* embryos were staged to NF stage 3 (4 cell). Using a microinjector, the two dorsally-fated blastomeres were both injected with 2 nL of morpholino mix. Morpholino mix contained a morpholino against *meis1* (MO1:8 ng or MO2:16 ng) or against *pbx3* (MO1:10 ng or MO2:10 ng) and a labeled dextran tracer (used at 2 mg/mL, ThermoFisher D1817). Embryos were screened at stage 18 for bilateral red fluorescence in the nervous system tissues. Neurulas were raised to stage 41 and used for regeneration assays. Morphant regenerates were collected at 24hpa, 48hpa, and 72hpa for measurement analysis.

### Tail vein injections

Wild-type *X. tropicalis* embryos were staged to NF embryonic stage 35–36 (pre-hatching stage). Tadpoles were anaesthetized with MS-222 and moved from culture dish to an agarose coated dish in one drop. Surrounding media was removed. Using a microinjector, a pulled needle containing VIVO-morpholino and labeled dextran tracer was inserted into the ventral tail vein in the posterior tail and 3 × 2 nL injections were delivered (20 ng). Embryos were returned to fresh media and screened for tracer fluorescence in the blood stream. Injected tadpoles were grown 24 hr to stage 41 and amputated. At 24hpa, tadpoles to be collected at 72hpa were re-injected into the tail vein as before.

## CRISPR-mediated mutation

### sgRNA Design, Preparation, and Injection

sgRNAs were designed targeting meis1 and pbx3 using CRISPRSCAN (https://www.crisprscan.org/?page=gene) (options: organism = 'Frog-*Xenopus tropicalis*', Gene = 'pbx3' or 'meis1', Cas9-NGG, In Vitro T7 promoter). PCR was performed as described in *Bhattacharya et al. (2015)*. SgRNA was transcribed using T7 mMachine kit (Ambion). Guides were injected into 2 blastomeres of 4 cell embryos with 1.5 ng Cas9 (*Bhattacharya et al., 2015*; *Nakayama et al., 2013*). All guides were injected at a dose of 400 pg/embryo.

## Immunohistochemistry

### Whole-mount

*X. tropicalis* tadpoles were fixed for 20 min in 10xMEMFA, 3.7% formaldehyde at room temperature. To preserve reporter fluorescence, tadpoles were kept dark upon fixing. Tadpoles were permeabilized by washing 3 × 20 min in PBS + 0.01% Triton x-100 (PBT). Tadpoles were blocked for 1 hr at room temperature in 10% CASblock (Invitrogen #00–8120) in PBT. Then tadpoles were incubated in primary antibody (1:50 neurofilament associated antigen, DSHB 3A10; 1:1000 Anti-Histone H3 (tri methly K9, Phospho S10), Abcam 14955) diluted in 100% CAS-block overnight at 4°C. Tadpoles were then washed 3 × 10 min at room temperature in PBT and reblocked for 30 min in 10% CAS-block in PBT. Secondary antibody (goat anti-mouse 488, ThermoFisher A11001 or goat anti-mouse 555, ThermoFisher A21422) were diluted 1:500 in 100% CAS-block and incubated for 2 hr at room temperature.

## Sections

Sections were baked onto slides on a heat block for 1 hr. Once slide had cooled, a hydrophobic barrier was drawn around the sections with a PAP pen (cat) and left to dry. Once dry, sections were washed with PBT 3 × 10 min. Sections were then blocked for 1 hr in 10% CASBlock followed by incubation in primary antibody overnight at 4C (1:50 neurofilament associated antigen, DSHB 3A10; 1:100 anti-DCX, Cell Signaling Technology 4604S; 1:200 anti-Sox2, Cell Signaling Technology 2748S). The following day, primary antibody was removed and replaced with a brief wash in PBT. Sections were blocked for 30 min followed by 2 hr incubation in secondary antibody at room temperature (1:500 goat anti-mouse 555, ThermoFisher A21422; 1:500 goat anti-rabbit 594, ThermoFisher Cat. A-21422).

For both whole-mount and sections, tadpoles were then washed 3 × 20 min in PBT. Tadpoles were then incubated in 1:2000 DAPI (Sigma D9542) for 10 min before being washed with 1xPBS for 10 more minutes. Isolated tails were mounted on slides in ProLong Gold (ThermoFisher P36930). Images were acquired with a Leica SP8 confocal microscope.

## Regenerated tissue measurement and analysis

### Measurement

To measure regenerated length of spinal cord and tail, regenerated tails were mounted on slides and were imaged on the Images were acquired with a Lecia DM 5500 B microscope with the 10x or 20x objective. Using the Leica imaging software (LasX), measurements were taken from the amputation plane to the most posterior point of the axial tissues for tail length and amputation plane to most posterior point of the spinal cord for spinal cord length. We used morphological features such as disruption to somite chevron morphology to identify the amputation plane.

### Analysis

We used the R package ggplot2 to plot boxplots of the regenerate measurement data. A single regenerated tadpole was treated as one measurement. A two-tailed t-test was used to test how different the uninjected controls were from the morphants. Statistical analysis was performed using the R package ggpubr.

## PH3 cell quantification

24hpa and 72hpa regenerated tails stained with anti-PH3 and DAPI were imaged on a Leica DM 5500 B Microscope with a 20x objective. Z-Stacks were acquired encompassing the entire spinal cord in focus, while excluding the most outer structures (i.e. epidermis) and max projected. In FIJI, the regenerated spinal cord in was identified and outlined in the DAPI channel to create a region of interest. The region of interest was transferred to the PH3 channel, and the area of the region was taken. PH3+ cells were counted manually within the region of interest. Quantifications were reported as PH3+ cells per spinal cord area. Significance was tested by using a two tailed t-test between conditions in question. This test was performed with the R package ggpubr.

## Acknowledgements

We thank members of the Wills lab for critical comments during the preparation of this manuscript and support in frog husbandry. We thank Rosalind Bump for the 72hpa *ass1* in situ hybridization image. We thank members of MF4 Supergroup and Regeneration Club for productive discussion and helpful suggestions throughout the course of this project. We thank the UW Pathology Flow Cytometry Core Facility for help with FACS. We thank the Reh lab for access and training on their cryostat. We thank Xenbase for curation of genomic and literature information and the National *Xenopus* Resource for frogs. ADK was supported by the Cellular and Molecular Biology Training Grant PHS NRSA T32GM007270 from NIGMS. This work was supported by NINDS R01NS099124 to AEW.

## Additional information

### Funding

| Funder | Grant reference number | Author |
|---|---|---|
| National Institute of Neurological Disorders and Stroke | R01NS099124 | Andrea Elizabeth Wills |
| National Institute of General Medical Sciences | T32GM007270 | Anneke Dixie Kakebeen |

The funders had no role in study design, data collection and interpretation, or the decision to submit the work for publication.

### Author contributions
Anneke Dixie Kakebeen, Conceptualization, Data curation, Software, Formal analysis, Funding acquisition, Investigation, Visualization, Methodology, Project administration; Alexander Daniel Chitsazan, Software, Formal analysis; Madison Corinne Williams, Data curation, Visualization; Lauren M Saunders, Data curation, Formal analysis; Andrea Elizabeth Wills, Conceptualization, Resources, Data curation, Formal analysis, Supervision, Funding acquisition, Visualization, Methodology

### Author ORCIDs
Anneke Dixie Kakebeen (iD) https://orcid.org/0000-0002-4268-3028
Lauren M Saunders (iD) https://orcid.org/0000-0003-4377-4252
Andrea Elizabeth Wills (iD) https://orcid.org/0000-0003-3647-8105

### Ethics
Animal experimentation: This study was performed in strict accordance with the recommendations in the Guide for the Care and Use of Laboratory Animals of the National Institutes of Health. All of the animals were handled according to approved institutional animal care and use committee (IACUC) protocols (#4374) of the University of Washington, an AALAC-accredited institution. All surgeries were carried out under MS222 or Benzocaine anesthesia, and every effort was made to minimize suffering.

### Decision letter and Author response
Decision letter https://doi.org/10.7554/eLife.52648.sa1
Author response https://doi.org/10.7554/eLife.52648.sa2

## Additional files

### Supplementary files
• Supplementary file 1. Supplementary output tables. (**a**) ATAC-Seq sample preparation details. (**b**) ATAC-Seq quality control metrics. (**c**) Pax6 vs. all Tissue gene ontology results (more accessible in pax6 libraries). (**d**) Pax6 vs. all Tissue gene ontology results (more accessible in all-tissue libraries). (**e**) 6hpa gene ontology results. (**f**) 24hpa gene ontology results. (**g**) 72hpa gene ontology(**h**) 6hpa ReviGO results. (**i**) 24hpa ReviGo results. (**j**) 72hpa ReviGo results. Key Resource Table. Reagents table.

• Supplementary file 2. Key Resources Table.

• Transparent reporting form

### Data availability
Sequencing data has been deposited in GEO under accession code GSE146837 (https://www.ncbi.nlm.nih.gov/geo/query/acc.cgi?acc=GSE146837).

The following datasets were generated:

| Author(s) | Year | Dataset title | Dataset URL | Database and Identifier |
|-----------|------|---------------|-------------|-------------------------|
| Kakebeen A, Chitsazan A, Williams M, Saunders L, Wills A | 2020 | Chromatin accessibility dynamics and single cell RNA-Seq reveal new regulators of regeneration in neural progenitors | http://www.ncbi.nlm.nih.gov/geo/query/acc.cgi?acc=GSE146830 | NCBI Gene Expression Omnibus, GSE146830 |
| Kakebeen A, Chitsazan A, Williams M, Saunders L, Wills A | 2019 | Chromatin accessibility dynamics and single cell RNA-Seq reveal new regulators of regeneration in neural progenitors | http://www.ncbi.nlm.nih.gov/geo/query/acc.cgi?acc=GSE146836 | NCBI Gene Expression Omnibus, GSE146836 |
| Kakebeen A, Chitsazan A, Williams M, Saunders L, Wills A | 2020 | Chromatin accessibility dynamics and single cell RNA-Seq reveal new regulators of regeneration in neural progenitors | https://www.ncbi.nlm.nih.gov/geo/query/acc.cgi?acc=GSE146837 | NCBI Gene Expression Omnibus, GSE146837 |

The following previously published dataset was used:

| Author(s) | Year | Dataset title | Dataset URL | Database and Identifier |
|-----------|------|---------------|-------------|-------------------------|
| Chang J, Baker J, Wills A | 2017 | RNA-Seq of Xenopus tail regeneration | https://www.ncbi.nlm.nih.gov/geo/query/acc.cgi?acc=GSE88975 | NCBI Gene Expression Omnibus, GSE88975 |

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
