## [Decision Letter]

**Acceptance summary:**

This manuscript investigates the transcriptional changes in neural tissue accompanying tail regeneration in *Xenopus tropicalis* tadpoles. The key findings are that (1) the initial transcriptional response promotes neuronal differentiation, followed within 3 days by an increase in expression of genes associated with cell proliferation; and (2) meis1 and pbx3 emerge as important regulators of the neural regenerative response. This first finding in particular is novel and challenges the fundamental assumption that cell dedifferentiation and proliferation are the initial steps in regeneration. The novelty and potential significance of these observations, together with the overall quality of the work, merit publication in *eLife*. Moreover, the manuscript is very clearly presented, and the logical flow of the computational analyses is easy to follow.

**Decision letter after peer review:**

Thank you for sending your article entitled "Chromatin accessibility dynamics and single cell transcriptomics reveal new regulators of neural progenitor regeneration" for peer review at *eLife*. Your article is being evaluated by three peer reviewers, and the evaluation is being overseen by a Reviewing Editor and Marianne Bronner as the Senior Editor.

All of the reviewers find the manuscript interesting and think that it will be potentially very useful to the community. That said, they also raise substantial concerns that will require major revisions.

Reviewer #1:

The manuscript by Kakebeen et al., is addressing a fundamental question in developmental and regenerative biology. With the realization of human therapies for nervous system regeneration after spinal cord injury, it is important to understand the strategies for regeneration that nature has already solved. The advent of new techniques such as scRNAseq and ATAC-seq have generated opportunities to study changes in cell state and type in complex cell populations, which Kakebeen et al., has made progress with here. Considering the authors are addressing a classic question in regenerative biology using the latest techniques, they’re of great value to the study. Overall, the ATACseq and scRNAseq seem robust, but there are some significant flaws in the design of the study that limit the interpretation that can be made by the readers. These problems are described below.

1) The choice of *pax6:GFP* as the driver instead of *Sox2* is a major drawback of the paper, unless major reinterpretation of the data is performed. The impression made from the Introduction and Results is that the study focuses on all NPCs. Based upon previous literature, this is likely not the case. Pax6 plays a well-established role in determining the dorsal/ventral fate of NSCs in the developing spinal cord across vertebrates, namely the pMN domain, and is expressed in a restricted intermediate dorsal/ventral zone of the vertebrate spinal cord (Pituello, 1997; Reimer et al., 2009). The image in Figure 1B needs to be of higher quality in order to determine the expression pattern of the transgene, but it looks to be and is most likely expressed in a canonical fashion in the intermediate D/V zone of the spinal cord. I would suggest a thinner cryosection and a zoom in on the spinal cord specifically. Pax6 has also been associated with controlling cell cycle exit in neural progenitor cells towards differentiation (Bel-Vialar, Medevielle, and Pituello, 2007; Osumi et al., 2008), which should be kept in mind throughout the study. Also, canonical D/V pax6 expression is maintained in mature axolotl salamander spinal cords and dorsal/ventral position of NSCs mainly remain stable after spinal cord regeneration in the axolotl (McHedlishvili et al., 2007). Considering all interpretation of the manuscript is under the assumption that all NPCs are marked and differentiation is not specific to a D/V region, the interpretation is missing a lot of likely very interesting findings on pax6 function and D/V patterning during spinal cord regeneration. As is, it is difficult to interpret exactly what the cell population the *pax6:GFP* cells are labeling and what they are differentiating into during regeneration (see next statement).

2) It should be explained why cre/lox lineage tracing was not used in the study. As presented, it is unclear if the cell population that is being sorted is exclusively the NSCs or does it include the neurons they differentiate into. Based upon the gene families that were identified from ATACseq including genes involved in neuritogenesis and growth cone formation, it is likely the authors also sorted differentiated neurons or cells in the process of differentiating. It is known that GFP protein can last for days long after the transgene is not expressed. Use of a *Sox2* Cre driver *Xenopus* mated with a fluorescent cre reporter would have overcome the ambiguity of the cells that are being studied and provide a clear trajectory of NSC to differentiated cell types.

3) The in situ hybridizations in Figure 5 do not support the sequencing data. At stage 41, both genes look to be expressed in all mesodermal tissue such as muscle. In subsection “Gene regulatory network prediction reveals pbx3 and meis1 as candidate

regulators of neuronal regeneration”, it states "Of these, pbx3 was the most restricted to the neural lineage (Supp. 3 E/G)". This is emphasized again in the Discussion. The in situ hybridization shows pbx3 is expressed throughout the mesenchyme of the regenerating tail at 48dpa. It will be important to perform cross sections of tissues to show expression in the expected domains, which will likely be the intermediate DV regions of the spinal cord (see new interpretations of pax6 cells above). I would suggest commercially available FISH such as hybridization chain reaction (molecularinstruments.com) or RNAscope that has higher resolution in the cellular level.

Reviewer #2:

The manuscript by Kakebeen et al., ("Chromatin accessibility dynamics and single cell transcriptomics reveal new regulators of neural progenitor regeneration") investigates the transcriptional changes in neural tissue accompanying tail regeneration in *Xenopus tropicalis* tadpoles. The key findings are that (1) the initial transcriptional response promotes neuronal differentiation, followed within 3 days by an increase in expression of genes associated with cell proliferation; and (2) meis1 and pbx3 emerge as important regulators of the neural regenerative response. This first finding in particular is novel and challenges the fundamental assumption that cell dedifferentiation and proliferation are the initial steps in regeneration. The novelty and potential significance of these observations, together with the overall quality of the work, merit publication in *eLife*. Moreover, the manuscript is very clearly presented, and the logical flow of the computational analyses is easy to follow.

I have two points of concern. The first is that, while the authors claim that pbx3 is predicted to be a key transcriptional regulator of the neural regenerative response, the results in Figure 5 indicate that only a very small proportion of neural cells express pbx3. (I found Figure 5E-5F somewhat confusing, and it may be that clarification of the figure legends will address this issue). If my understanding of these results is correct, however, then I suggest the authors revise the text, since, as written, it implies that pbx3 is critical to the entire regenerative response.

The second is in regard to the morpholino oligonucleotide (MO) experiments. The authors use two different MOs for each of the two targets, in lieu of a mispair control. While the pbx3 MOs lead to quite similar results, the meis1 MOs produce somewhat distinct phenotypes (compare Figure 5 C-D, 5W-X). This may be a sample size issue, or it may arise from the apparent greater effectiveness of MO1 vs MO2. Given the ongoing discussion regarding MO use, I would like to see the authors address this point in some way, either experimentally (e.g., a western blot comparing effectiveness of the two MOs in regulating endogenous pbx3 [Abcam has an antibody that works for mammalian species and zebrafish], but other strategies would be suitable) or via discussion in the text, which might also include results of computational comparisons to identify any other predicted targets.

Reviewer #3:

In this manuscript, Kakebeen et al. analyzed chromatin accessibility and single cell transcriptomes from pax6-driven GFP positive cells during *Xenopus* tail regeneration. The authors utilized transgenic *pax6:GFP* to sort neural progenitor cells (NPCs) throughout this study. They used deep sequencing, ATACseq, and SCseq to define transcripts, biological processes, and transcription factors that are central to spinal cord regeneration in the regenerating *Xenopus* tail. The authors concluded that NPCs place an early priority on neuronal differentiation early after injury, and prioritize proliferation at later stages of regeneration. They go on to perform morpholino-based functional analysis on two candidate transcription factors, and concluded that meis1 and pbx3 are required for spinal cord development and regeneration.

Overall, bulk RNA-seq, ATACseq and SCseq offer a powerful combination of tools to infer new insights into the cellular and temporal regulation of endogenous spinal cord regeneration *Xenopus*. The battery of analyses performed on these datasets is well-executed, revealing an early emphasis (6 hpa) on neuronal differentiation and a later emphasis on neural progenitor replenishment (72 hpa). The authors also use their scRNA-seq dataset to define new markers for neural cell types. However, this hypothesized temporal regulation of NPC injury responses is weakened by the following concerns:

1) The entire study is based on the assumption that pax6 is a general marker for NPCs prior to and during regeneration, and that *pax6:GFP* recapitulates endogenous pax6 expression throughout the time course of regeneration. While all the conclusions from this study are based on these assumptions, *pax6:GFP* transgenic animals were poorly characterized (Figure 1 and Figure 1—figure supplement 1).

– Higher magnification images and extensive co-labeling with well-established NPC markers throughout regeneration are critical for this study.

– Confirming that the transgene recapitulates endogenous pax6 expression is also critical for the study. The authors state *pax6:GFP* is consistent with developmental expression of pax6, but this is assuming regeneration will fully recapitulate development, which is not always the case. An additional concern is the stability of GFP over fine time windows as narrow as 6 hours.

– From the low mag images in Figure 1D, it seems that the transgene is not restricted to the spinal cord. This observation is further supported by scRNA-seq (Figure S3A), which indicates that pax6 is expressed in many cell types beyond NPCs.

2) Even with a full characterization of the transgene, this reviewer is always concerned about drawing conclusions based on a single genetic tool. In this case, it is impossible to rule out the presence of pax6 negative progenitor cells that differentially contribute proliferation versus differentiation at different time points after injury. Further, equally impossible is to rule out technical cell dissociation limitations that may cause differential cell representations in the analysis.

3) The main premise of the study is that NPCs place an early priority on neuronal differentiation before prioritizing proliferation. This is an interesting and potentially relevant observation that warrants experimental support.

– Detailed elucidation of cell proliferation (rostral and caudal to the lesion), migration, and differentiation are recommended in this case.

– These studies would also address the possibility that proliferation is occurring rostral to lesion and that NPCs are recruited into the regenerate where they differentiate into neurons.

– The authors supported their conclusion that NPC proliferation is not abundant until 72 hpa by referencing Love et al., 2011 and their own work (Chang et al., 2017). Chang et al., 2017 does not mention cell proliferation and instead shows an emphasis on immune response at 72 hpa in whole tail. Love et al., 2011 does show an increase in cell proliferation in whole tail at 72 hpa. To support the authors' conclusion that this boost in cell proliferation is indeed happening in the spinal cord, a marker for cell proliferation should be used in the *pax6:GFP* expressing tadpoles following tail transection.

4) The authors followed up on their SCseq findings by performing functional analyses for meis1 and pbx3. Unfortunately, these studies present a number of major concerns that dampen the excitement about the study.

– The morpholino experiments do not follow recent guidelines for morpholino use. Proper controls are recommended in this case.

– It is unclear how the authors concluded CNS/spinal cord developmental defects based on the severe morpholino phenotypes that were obtained even at the lower doses. These phenotypes are consistent with generic, overall toxicity phenotypes that are common to morpholinos.

– The severe developmental phenotypes caused by meis1 and pbx3 cloud the interpretation of regeneration phenotypes in these morphants.

– Neurofilament stains for uninjured wildtype and morphant animals should be included in Figure 6.

– What is the effect of meis1 and pbx3 morpholinos on cell proliferation in both injured and uninjured tadpoles?

– The authors claim meis1 is required for the neuronal differentiation occurring at early timepoints during spinal cord regeneration. They show that axons in the regenerating tail are disorganized in meis1 morphants, but are these neurons the appropriate neuron type?

[Editors' note: further revisions were suggested prior to acceptance, as described below.]

Thank you for submitting your article "Chromatin accessibility dynamics and single cell RNA-Seq reveal new regulators of regeneration in neural progenitors" for consideration by *eLife*. Your article has been reviewed by three peer reviewers, and the evaluation has been overseen by a Reviewing Editor and Marianne Bronner as the Senior Editor The following individuals involved in review of your submission have agreed to reveal their identity: James R Monaghan (Reviewer #1); Amy Sater (Reviewer #2).

The reviewers have discussed the reviews with one another and the Reviewing Editor has drafted this decision to help you prepare a revised submission.

Summary:

This revised manuscript has addressed most of the concerns of the reviewers and is fundamentally suitable for publication after some edits in response to the few remaining issues raised by the reviewer. I ask you to address these to the best of your ability with editorial changes.

Revisions:

1) Subsection “Meis1 and Pbx3 are necessary for successful spinal cord and tail

regeneration” first paragraph: please add "antibody" after "anti-neurofilament"

2) The addition of *Sox2* antibody staining and its overlap with *pax6:GFP* is informative. There is a small number of Sox2^+^ cells that are *pax6:GFP* negative, which the authors address in the text. However, reviewer 3 has two concerns with this data:

a) *Sox2*/*pax6:GFP* co-labelling is performed during tadpole development. This approach assumes *pax6:GFP* and *Sox2* co-expression behave similarly in regeneration as in development, which is not necessarily the case.

b) The authors do not address pax6:GFP+ Sox2- cells. These cells, which are observed by histology and scRNA-seq analysis, could affect data interpretation in Figures 2 and 3.

3) pax6:GFP- Sox2^+^ NPCs do exist. How are these cells contributing to the differences seen in cell proliferation and differentiation during different stages of regeneration? Perhaps these are the pH3+ cells present at 24 hpa. These pax6:GFP- Sox2^+^ cells appear to be fewer in number, but their presence should be more clearly acknowledged in the text.

---

## [Author Response]

Reviewer #1:1) The choice of pax6:GFP as the driver instead of Sox2 is a major drawback of the paper, unless major reinterpretation of the data is performed. The impression made from the Introduction and Results is that the study focuses on all NPCs. Based upon previous literature, this is likely not the case. Pax6 plays a well-established role in determining the dorsal/ventral fate of NSCs in the developing spinal cord across vertebrates, namely the pMN domain, and is expressed in a restricted intermediate dorsal/ventral zone of the vertebrate spinal cord (Pituello, 1997; Reimer et al., 2009). The image in Figure 1B needs to be of higher quality in order to determine the expression pattern of the transgene, but it looks to be and is most likely expressed in a canonical fashion in the intermediate D/V zone of the spinal cord. I would suggest a thinner cryosection and a zoom in on the spinal cord specifically. Pax6 has also been associated with controlling cell cycle exit in neural progenitor cells towards differentiation (Bel-Vialar, Medevielle, and Pituello, 2007; Osumi et al., 2008), which should be kept in mind throughout the study. Also, canonical D/V pax6 expression is maintained in mature axolotl salamander spinal cords and dorsal/ventral position of NSCs mainly remain stable after spinal cord regeneration in the axolotl (McHedlishvili et al., 2007). Considering all interpretation of the manuscript is under the assumption that all NPCs are marked and differentiation is not specific to a D/V region, the interpretation is missing a lot of likely very interesting findings on pax6 function and D/V patterning during spinal cord regeneration. As is, it is difficult to interpret exactly what the cell population the pax6:GFP cells are labeling and what they are differentiating into during regeneration (see next statement).

We have added new cryosections of the posterior spinal cord at stage 41 to Figure 1, which more clearly show the domain of *pax6:GFP* expression (Figure 1C), as well as the overlap between this domain and *Sox2* protein expression (Figure 1D). These have highlighted that there is broad overlap between the GFP domain (which surrounds the central canal) and the *Sox2* domain. We note that some GFP positive cells can be detected across the D/V axis of the spinal cord, and that while GFP expression is weaker in the dorsal-most potion of the spinal cord, there are also few Sox2^+^ cells that we could detect in this region and at this stage. Recognizing that the overlap is imperfect, we also have taken care to note in the text (both in the results for Figure 1 and in the discussion), that we may not have included every NPC in our FACS analysis, and to more precisely state that our findings apply to pax6+ NPCs.

2) It should be explained why cre/lox lineage tracing was not used in the study. As presented, it is unclear if the cell population that is being sorted is exclusively the NSCs or does it include the neurons they differentiate into. Based upon the gene families that were identified from ATACseq including genes involved in neuritogenesis and growth cone formation, it is likely the authors also sorted differentiated neurons or cells in the process of differentiating. It is known that GFP protein can last for days long after the transgene is not expressed. Use of a Sox2 Cre driver *Xenopus* mated with a fluorescent cre reporter would have overcome the ambiguity of the cells that are being studied and provide a clear trajectory of NSC to differentiated cell types.

While we agree a *Sox2* Cre driver would be a powerful tool, there is not currently a line resembling this available in *X. tropicalis*. Given the 6-8 month generation time of *X. tropicalis* in our own breeding colony and the 3-month interval required between ovulations of each potential founder female, as well as the approximately $25k in sequencing costs that would be required to recapitulate the study, creating these data anew was beyond the scope of what we could take on in this study. However, we have added high-resolution cryosections comparing the *pax6:GFP* domain with *Sox2* (NSC), Dcx (neurons) and neurofilament stains to better characterize the populations labeled (Figure 1D-F). These show that cytoplasmic GFP expression is in more medial cells of the spinal cord, as are the Sox2^+^ nuclei, while neuronal markers localize more the spinal cord periphery. We have highlighted in the Results sections that while the domain of GFP expression accords best with *Sox2*, not every NPC may be captured in our study, and some neurons may be have been included (Results first paragraph). The population-level conclusions in our ATAC-Seq data are not changed, especially when paired by the analysis of specific neural cell population dynamics we present through scRNA-Seq analysis.

3) The in situ hybridizations in Figure 5 do not support the sequencing data. At stage 41, both genes look to be expressed in all mesodermal tissue such as muscle. In subsection “Gene regulatory network prediction reveals pbx3 and meis1 as candidateregulators of neuronal regeneration”, it states "Of these, pbx3 was the most restricted to the neural lineage (Supp. 3 E/G)". This is emphasized again in the Discussion. The in situ hybridization shows pbx3 is expressed throughout the mesenchyme of the regenerating tail at 48dpa. It will be important to perform cross sections of tissues to show expression in the expected domains, which will likely be the intermediate DV regions of the spinal cord (see new interpretations of pax6 cells above). I would suggest commercially available FISH such as hybridization chain reaction (molecularinstruments.com) or RNAscope that has higher resolution in the cellular level.

We have provided new *in situs* for *meis1* and *pbx3* in the new version of the figure (now Figure 6), and in particular have included cryosections to better show the spinal cord expression pattern in the tail, which spans a broad D/V domain (Figure 6G-J). A challenge of *in situs* for both factors is the very low overall level of their expression (which is likely a contributing factor to why they have not previously been identified in regeneration!) We think the new sections better show their expression in the uninjured context. Expression of both factors increases following injury, and the *in situs* at 24hpa and 72hpa more clearly show their localization in the spinal cord (Figure 6K-P).

Reviewer #2:I have two points of concern. The first is that, while the authors claim that pbx3 is predicted to be a key transcriptional regulator of the neural regenerative response, the results in Figure 5 indicate that only a very small proportion of neural cells express pbx3. (I found Figure 5E-5F somewhat confusing, and it may be that clarification of the figure legends will address this issue). If my understanding of these results is correct, however, then I suggest the authors revise the text, since, as written, it implies that pbx3 is critical to the entire regenerative response.

We have edited the Results section and the figure legend for Figure 5 (now Figure 6) to better clarify the expression of these factors. Indeed, it is a relatively small proportion of neural cells that contain detectable levels of transcripts for these factors (it’s worth noting here that a larger proportion of cells may have some degree of expression, as scRNA-Seq captures only a fraction of transcripts for each cell). Because *pbx3* and *meis1* are expressed at low transcriptional levels and in only a small fraction of cells, we also find it impressive that they are both critical to a more extensive regenerative response. In the discussion, we speculate that this may be tied to the phenomenon of neural dependence, which has been described previously in multiple species, in which functional nerves are required for the ability of other tissue types to regenerate.

The second is in regard to the morpholino oligonucleotide (MO) experiments. The authors use two different MOs for each of the two targets, in lieu of a mispair control. While the pbx3 MOs lead to quite similar results, the meis1 MOs produce somewhat distinct phenotypes (compare Figure 5 C-D, 5W-X). This may be a sample size issue, or it may arise from the apparent greater effectiveness of MO1 vs MO2. Given the ongoing discussion regarding MO use, I would like to see the authors address this point in some way, either experimentally (e.g., a western blot comparing effectiveness of the two MOs in regulating endogenous pbx3 [Abcam has an antibody that works for mammalian species and zebrafish], but other strategies would be suitable) or via discussion in the text, which might also include results of computational comparisons to identify any other predicted targets.

a) We chose more representative images for *meis1* MOs, which better show the concordance of phenotypes between the two MOs (Figure 7—figure supplement 1C/E). We also noted that no off-target binding was predicted for any of the MOs used in the Materials and methods section.

b) We have added to our morpholino experiments by designing CRISPR guides for both *meis1* and *pbx3*. These guide RNAs phenocopied the morpholino results when injected with Cas9 protein (Figure 7—figure supplement 1G-N)

c) We attempted Western blot experiments for both Meis1 and Pbx3, but unfortunately were unable to see sufficiently strong bands for either protein to evaluate the degree of knockdown. This is likely because of the low and tissue-specific expression of both factors.

d) Finally, we used stage 41 tail vein injection of vivo-MO versions of *pbx3* and *meis1* MOs to bypass early embryonic effects (new Figure 7) (see responses to reviewer 3).

Reviewer #3:1) The entire study is based on the assumption that pax6 is a general marker for NPCs prior to and during regeneration, and that pax6:GFP recapitulates endogenous pax6 expression throughout the time course of regeneration. While all the conclusions from this study are based on these assumptions, pax6:GFP transgenic animals were poorly characterized (Figure 1 and Figure 1—figure supplement 1).– Higher magnification images and extensive co-labeling with well-established NPC markers throughout regeneration are critical for this study.– Confirming that the transgene recapitulates endogenous pax6 expression is also critical for the study. The authors state pax6:GFP is consistent with developmental expression of pax6, but this is assuming regeneration will fully recapitulate development, which is not always the case. An additional concern is the stability of GFP over fine time windows as narrow as 6 hours.– From the low mag images in Figure 1D, it seems that the transgene is not restricted to the spinal cord. This observation is further supported by scRNA-seq (Figure S3A), which indicates that pax6 is expressed in many cell types beyond NPCs.

a) We have added higher-resolution cross-sectional images of the GFP domain, as well as co-labeling with *Sox2*, Dcx and neurofilament to Figure 1C-F. These show that *pax6:GFP* labels cells in a broad D/V domain that surrounds the central canal, much like *Sox2*, while differentiated neuronal markers (Dcx, neurofilament) label the periphery. In recognition of the possibility that we have not captured every NSC, or may have captured some differentiated neurons, we added additional text to the Results and Discussion sections (see also response to reviewer 1).

b) Our initial imaging of the transgene was imperfect due to autofluorescence and bleed-through from the cardiac actin:RFP label; new images better show that expression is neural-specific.

2) Even with a full characterization of the transgene, this reviewer is always concerned about drawing conclusions based on a single genetic tool. In this case, it is impossible to rule out the presence of pax6 negative progenitor cells that differentially contribute proliferation versus differentiation at different time points after injury. Further, equally impossible is to rule out technical cell dissociation limitations that may cause differential cell representations in the analysis.

These are important caveats to consider, which we have addressed in the revised Results sections.

3) The main premise of the study is that NPCs place an early priority on neuronal differentiation before prioritizing proliferation. This is an interesting and potentially relevant observation that warrants experimental support.– Detailed elucidation of cell proliferation (rostral and caudal to the lesion), migration, and differentiation are recommended in this case.– These studies would also address the possibility that proliferation is occurring rostral to lesion and that NPCs are recruited into the regenerate where they differentiate into neurons.– The authors supported their conclusion that NPC proliferation is not abundant until 72 hpa by referencing Love et al., 2011 and their own work (Chang et al., 2017). Chang et al., 2017 does not mention cell proliferation and instead shows an emphasis on immune response at 72 hpa in whole tail. Love et al., 2011 does show an increase in cell proliferation in whole tail at 72 hpa. To support the authors' conclusion that this boost in cell proliferation is indeed happening in the spinal cord, a marker for cell proliferation should be used in the pax6:GFP expressing tadpoles following tail transection.

We have generated a new data figure, Figure 5, that addresses these points.

a) We addressed the question of proliferation by adding quantification of PH3-positive cells to new Figure 5 Y-AA. Because the cell density in the spinal cord is low, PH3-positive cells were rarely captured in transverse sections, and so we used high resolution imaging to capture optical longitudinal sections instead, and looked for overlap between PH3-positive cells and the GFP domain (Materials and methods reflect accordingly). This analysis showed that mitotic cells are rare at 24hpa, and far more abundant at 72hpa (see particularly quantification data in new Figure 5AA). Although we did not have good tools to directly assay migration of proliferative cells in the spinal cord, we note that proliferative cells were rare in the spinal cord rostral to the injury site at either time point.

b) We also apologize strongly for the citation error, which has been corrected.

c) To address the question of neuronal differentiation, we elected to show the expression of several markers of differentiated neurons that our single-cell RNA-Seq identified. We chose 4 markers for this analysis, which all have strong expression in the regenerating neural ampulla at 24hpa (new Figure 5E, K, Q, W).

4) The authors followed up on their SCseq findings by performing functional analyses for meis1 and pbx3. Unfortunately, these studies present a number of major concerns that dampen the excitement about the study.– The morpholino experiments do not follow recent guidelines for morpholino use. Proper controls are recommended in this case.– It is unclear how the authors concluded CNS/spinal cord developmental defects based on the severe morpholino phenotypes that were obtained even at the lower doses. These phenotypes are consistent with generic, overall toxicity phenotypes that are common to morpholinos.– The severe developmental phenotypes caused by meis1 and pbx3 cloud the interpretation of regeneration phenotypes in these morphants.– Neurofilament stains for uninjured wildtype and morphant animals should be included in Figure 6.– What is the effect of meis1 and pbx3 morpholinos on cell proliferation in both injured and uninjured tadpoles?– The authors claim meis1 is required for the neuronal differentiation occurring at early timepoints during spinal cord regeneration. They show that axons in the regenerating tail are disorganized in meis1 morphants, but are these neurons the appropriate neuron type?

We have added to our MO experiments considerably, and have included a full new data figure (new Figure 7).

a) We were able to phenocopy the effects of the MOs by F0 CRISPR/Cas9 (Figure 7—figure supplement 1G-J)

b) We added neurofilament stains to the uninjured morphants as requested (Figure 7—figure supplement 2B-F)

c) Finally and most significantly, we agreed with the reviewer that the early embryonic defects clouded our ability to assess the roles of these two factors in tadpoles and specifically in regeneration. Therefore we had two of our MO sequences, one for pbx3 and one for meis1, resynthesized as tissue-permeable vivo-MOs. We optimized a tail-vein injection strategy to deliver these at late embryo stages and/or during regeneration. Both vivo-MOs cased a failure of the ventral intersomitic axons to properly elongate (new Figure 7C,D). During regeneration, these vivo-MOs resulted in reduced and mispatterned neurofilament staining in the regenerate when assayed at 72 hpa (new Figure 7E-G). Both vivo-MOs also resulted in reduced length of the tail and spinal cord, as well as a reduction in spinal cord area in the regenerate at 72hpa (new Figure 7K, L). (Results reported in the final paragraph of the Results section). We hope the reviewer will agree that this approach clarifies that Pbx3 and Meis1 are critical for proper regeneration, independent from their early embryonic functions, which we hope to further elucidate in a later study. As noted above in the response to Figure 2, we find it remarkable that the overall regeneration of the tail is impacted by the loss of function of these more specifically-expressed transcription factors, and speculate that this may be tied to the nerve dependence of regeneration, although a full interrogation of this connection is beyond the scope of the current paper.

[Editors' note: further revisions were suggested prior to acceptance, as described below.]

Revisions:1) Subsection “Meis1 and Pbx3 are necessary for successful spinal cord and tailregeneration” first paragraph: please add "antibody" after "anti-neurofilament"

Edit was made.

2) The addition of Sox2 antibody staining and its overlap with pax6:GFP is informative. There is a small number of Sox2+ cells that are pax6:GFP negative, which the authors address in the text. However, reviewer 3 has two concerns with this data:a) Sox2/pax6:GFP co-labelling is performed during tadpole development. This approach assumes pax6:GFP and Sox2 co-expression behave similarly in regeneration as in development, which is not necessarily the case.

We address that this is an assumption made using this method. We suggest that our scRNA-seq results point to NPCs continuing to express both markers, but also note that this analysis does not capture spatial dynamics in regeneration.

b) The authors do not address pax6:GFP+ Sox2- cells. These cells, which are observed by histology and scRNA-seq analysis, could affect data interpretation in Figures 2 and 3.

We address how we are likely capturing *pax6:GFP+* Sox2- cells. We note our hypothesis that these cells are derived from *pax6:GFP* neural progenitor cells but have since exited the progenitor state and still maintain GFP protein. We also suggest a way that we might address this point in the future.

3) pax6:GFP- Sox2+ NPCs do exist. How are these cells contributing to the differences seen in cell proliferation and differentiation during different stages of regeneration? Perhaps these are the pH3+ cells present at 24 hpa. These pax6:GFP- Sox2+ cells appear to be fewer in number, but their presence should be more clearly acknowledged in the text.

We address that we are capturing a broad selection of neural progenitors, however we are missing the dorsal most Sox2^+^ neural progenitor cell that are not also *pax6:GFP+.* We suggest follow up experiments that would address this concern in the future.